# Disrupted network interactions serve as a neural marker of dyslexia

Sabrina Turker [1,2,3 ✉], Philipp Kuhnke[1,2,3], Zhizhao Jiang[1] & Gesa Hartwigsen [1,2]

Dyslexia, a frequent learning disorder, is characterized by severe impairments in reading and writing and hypoactivation in reading regions in the left hemisphere. Despite decades of research, it remains unclear to date if observed behavioural deficits are caused by aberrant network interactions during reading and whether differences in functional activation and connectivity are directly related to reading performance. Here we provide a comprehensive characterization of reading-related brain connectivity in adults with and without dyslexia. We find disrupted functional coupling between hypoactive reading regions, especially between the left temporo-parietal and occipito-temporal cortices, and an extensive functional disruption of the right cerebellum in adults with dyslexia. Network analyses suggest that individuals with dyslexia process written stimuli via a dorsal decoding route and show stronger reading-related interaction with the right cerebellum. Moreover, increased connectivity within networks is linked to worse reading performance in dyslexia. Collectively, our results provide strong evidence for aberrant task-related connectivity as a neural marker for dyslexia that directly impacts behavioural performance. The observed differences in activation and connectivity suggest that one effective way to alleviate reading problems in dyslexia is through modulating interactions within the reading network with neurostimulation methods.

[1] Lise Meitner Research Group Cognition and Plasticity, Max Planck Institute for Human Cognitive and Brain Sciences, 04103 Leipzig, Germany. [2] Wilhelm Wundt Institute for Psychology, Leipzig University, 04103 Leipzig, Germany. [3]These authors contributed equally: Sabrina Turker, Philipp Kuhnke.
✉email: turker@cbs.mpg.de

Reading is a core feature of human communication and crucial for everyday social life, work, and interpersonal communication. From a cognitive perspective, the acquisition of fluent reading skills is based on multiple, hierarchically organized processes, including orthographic knowledge and recognition, orthographic-phonological mapping, and semantic access[1,2]. While known words are automatically accessed and retrieved as whole word forms[2], pseudowords are largely spelt out via grapheme-phoneme conversion[3]. The universal reading network in the human brain supporting these processes comprises three major circuits: (i) the left inferior frontal gyrus (IFG), (ii) the left dorsal temporo-parietal cortex (TPC), and (iii) the left ventral occipito-temporal cortex (vOTC)[4–6]. While the IFG is involved in attention, working memory and phonological output resolution[6], the TPC area is important for rule-based decoding and phonological processing[7]. The vOTC shows increasing sensitivity to print during reading acquisition and optimizes lexical access for reading[8].

5–10% of children worldwide fail to master literacy acquisition despite normal intelligence and cognitive functioning: they are diagnosed with dyslexia, a learning disability affecting reading and writing (ICD-11: 6A03.0[9])[10,11]. Symptoms usually persist into adulthood and impact the daily lives of affected individuals. Dyslexia is believed to result from a deficit in decoding and identifying the phonological properties of speech and/or accessing the respective phonological representations[12,13], but the exact underlying cause remains debated. At the neural level, children and adults with dyslexia fail to sufficiently engage the three core reading circuits during reading[11,14–16]. Specifically, activation of the left TPC is vital during early years[17] and has been found to predict response to interventions[18,19]. Longitudinal studies also suggest an atypically developing sensitivity to print in the left vOTC in dyslexia[8], which is linked to reading ability[20]. Apart from these classical areas, some studies provide evidence for the crucial roles of the thalamus[21] and the right cerebellum for reading[22] and for phonological processes[23].

Despite abundant evidence for marked hypoactivation in reading areas, there is little research on underlying changes in task-related network interactions in dyslexia. Consequently, it is unclear whether dyslexia is merely characterized by hypoactivation, dysfunctional network interactions, or both. Developmental studies suggest that phonological deficits are linked to impaired functional connectivity within the classical reading network[24,25], and aberrant connectivity with the visual network and domain-general networks[26]. In adults with dyslexia, differences in functional connectivity have been found between the left TPC and OTC during phonological assembly[4] and word reading[27]. Moreover, individuals with dyslexia show differences in functional coupling between the bilateral auditory cortices and left IFG during sound processing and the reading circuits and the thalamus during word and pseudoword reading[21,28]. In terms of task-related directed connectivity, studies in children with dyslexia reported decreased effective connectivity from and to the IFG, and increased connectivity with the OTC. Other small-sample studies with children found decreased connectivity with or within the OTC region during specific reading operations[29–31]. Yet, the number of studies is scarce and there is currently no clear picture of underlying neural network interactions in dyslexia, which would be mandatory to develop more effective interventions.

The present study was designed to fill this gap. We combined measures of task-related activation and functional as well as effective connectivity in adults with dyslexia and neurotypical readers. First, we aimed to identify key brain regions for overt simple and complex word and pseudoword reading. Second, and most importantly, we wanted to explore task-related connectivity between reading areas to unravel (impaired) interactions within the reading network. Third, we linked neural activation and connectivity to reading abilities to elucidate the behavioural relevance of such changes. To foreshadow our main results, we found strong behavioural differences that were accompanied by hypoactivation in the core reading areas and right-hemispheric regions in dyslexia. Functional connectivity analyses confirmed atypical coupling from several key reading areas to various brain regions in dyslexia, with the right cerebellum showing the largest disruption. Effective connectivity analyses suggest that adults with dyslexia process both words and pseudowords via a dorsal route, show more reading-related interactions with the right cerebellum, and have weaker but more distributed overall intrinsic coupling between reading regions. Furthermore, all neural parameters, including intrinsic connectivity both in the classical and extended reading network, were linked to reading performance. Overall, we provide strong evidence for aberrant functional and effective connectivity as a neural marker of dyslexia.

## Results

The present study investigated reading and spelling skills and the processing of simple and complex words and pseudowords during functional neuroimaging (fMRI) in typical readers and adults with dyslexia. We aimed to (1) detect potential differences in terms of functional activation during reading, (2) investigate functional connectivity between reading-relevant regions in typical and atypical readers, and (3) explore effective connectivity, i.e., directed coupling, within the reading network. Moreover, we investigated whether differences in functional activation and coupling were linked to behavioural performance in and outside the MRI.

**Reading and task performance.** Adults with dyslexia performed significantly worse on all administered reading, spelling and working memory tasks, except for digit span forward, nonword span and text reading accuracy (see Table 1). An illustration of the in-scanner task performance is provided in Fig. 1a.

Linear mixed models including the factors group (adults with dyslexia vs. control group), stimulus type (words vs. pseudowords) and complexity (simple vs. complex) showed that fMRI task performance also differed significantly between groups (Supplementary Tables 1–5; Fig. 1b). In line with the results of our model comparisons, a random intercept for subject was included in all three linear mixed models, but a random intercept for trial was only included in the model for speech onsets since the models did not converge. To summarize, adults with dyslexia had significantly longer speech onsets for all stimulus types. For reading times and accuracy, we found significant group differences for pseudowords, which were most pronounced for complex pseudowords.

**Functional activation.** The four conditions, simple and complex word and simple and complex pseudoword reading, recruited similar brain areas in both groups when compared against rest (see Supplementary Figs. 1–4). Univariate analyses further revealed distinct networks for words vs. pseudowords and complex vs. simple stimuli (Fig. 2). Pseudoword as compared to word reading relied on a largely bilateral network involving the bilateral vOTC, IFG/insula, posterior parietal cortex, and precentral gyrus. Activation was more pronounced during pseudoword reading in the left precentral gyrus, left IFG and left vOTC, as compared to their right homologues. In contrast, words as compared to pseudowords recruited a large network covering parts of the default mode network, such as the bilateral posterior TPC (angular gyri), bilateral

**Table 1 Descriptive subject data and group comparisons for behavioural scores.**

|  | CG | DYS |  |  |  |
|---|---|---|---|---|---|
|  | Mean (SD) | Mean (SD) | W-value | p-value | Cohen's d |
| Age | 28 (5) | 26.5 (6) | 418 | 0.323 | −0.272 |
| Nonverbal IQ | 107.6 (10.5) | 115.3 (11.4) | 199 | 0.028+ | 0.712 |
| Digit span forward | 8.3 (2.3) | 7.1 (1.9) | 421 | 0.068 | −0.568 |
| Digit span backward | 8.1 (2.1) | 5.9 (1.9) | 502 | <0.001*** | −1.098 |
| Nonword span | 137.1 (43.7) | 116.8 (32.6) | 427 | 0.179 | −0.525 |
| Pseudoword reading | 73.0 (18.1) | 45.9 (17.4) | 606 | <0.001*** | −1.526 |
| Word reading | 119.6 (17.5) | 76.0 (18.2) | 666 | <0.001*** | −2.443 |
| Spelling errors | 14.5 (6.2) | 35.5 (10.3) | 33 | <0.001*** | 2.482 |
| Text comprehension | 53.5 (7.7) | 45.9 (7.6) | 537 | <0.001*** | −0.993 |
| Text reading speed | 50.4 (8.2) | 40.4 (8.5) | 558 | <0.001*** | −1.186 |
| Text reading accuracy | 59.0 (11.3) | 60.0 (10.7) | 325 | 0.636 | 0.091 |
| Phoneme substitution | 4.5 (2.1) | 7.4 (3.6) | 142 | 0.003** | 0.989 |
| Letter naming (RAN) | 3.1 (0.5) | 2.4 (0.5) | 593 | <0.001*** | −1.4 |

Results comprise reading, spelling and working memory tasks. Values for the control group (CG; $n = 28$) and the dyslexia group (DYS, $n = 26$) indicate mean scores and standard deviations. Significant differences between groups were assessed with the Mann-Whitney $U$ test and $W$- and $p$-values, as well as effect sizes (Cohen's $d$) are reported.
CG control group, DYS adults with dyslexia, IQ intelligence quotient, RAN rapid automatized naming.
Significance levels: ***$p < 0.001$; **$p < 0.01$; *$p < 0.05$; + not significant after Bonferroni correction.

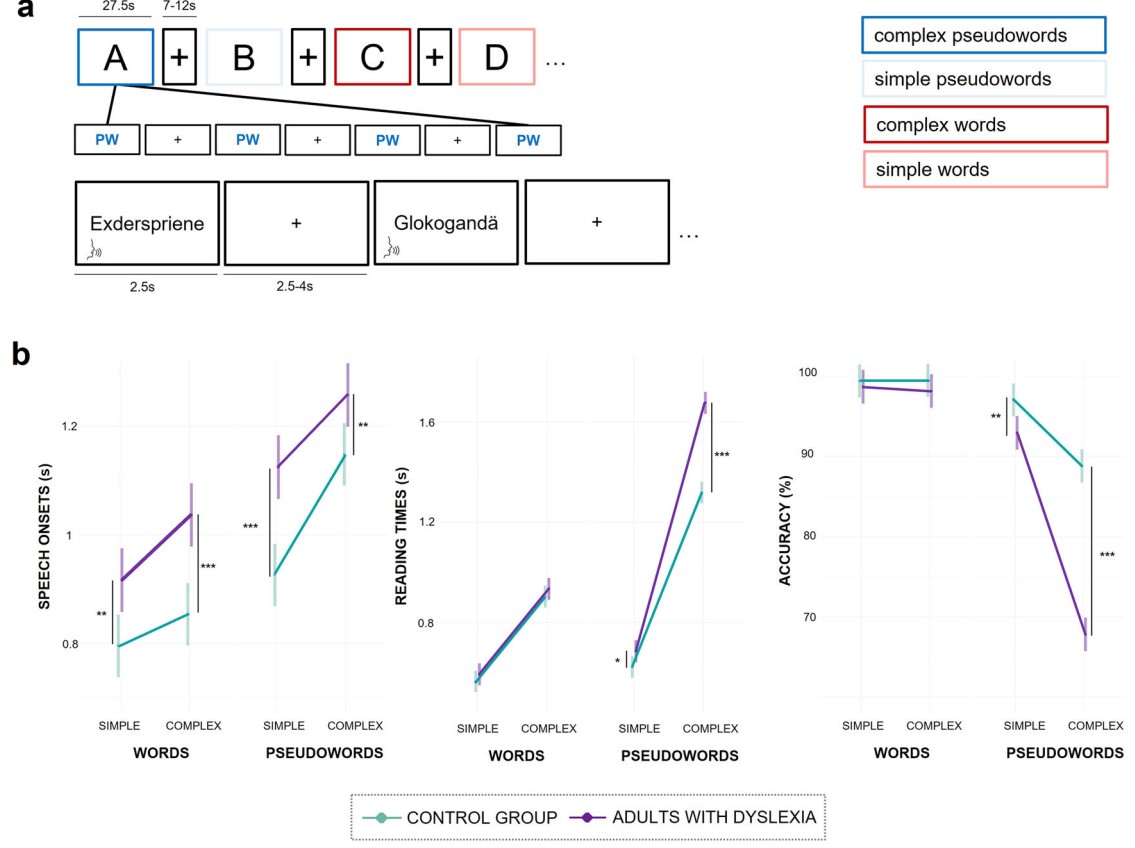

**Fig. 1 fMRI task design and differences in fMRI task performance in terms of speech onsets, reading times and reading accuracy. a** During fMRI, subjects read simple words (light red), complex words (dark red), simple pseudowords (light blue) and complex pseudowords (dark blue). Mini-blocks always contained 5 items and lasted for 27.5 s. Inter-stimulus intervals and rest periods were jittered (2.5–4 s and 7–12 s respectively). **b** Linear mixed model results show group differences in task performance for speech onsets, reading times and reading accuracy (control group: $n = 27$, adults with dyslexia $n = 26$). Significant post-hoc tests are indicated (***$p < 0.001$, **$p < 0.01$, *$p < 0.05$) and error bars (SE) are shown. Adults with dyslexia read all items slower but needed significantly longer to overtly produce pseudowords. Moreover, their accuracy was only significantly lower for the pseudoword reading blocks, with worse performance during complex pseudoword reading.

anterior temporal lobes and middle temporal gyrus (MTG), as well as superior frontal areas and medial areas.

Complex as compared to simple stimuli activated a bilateral occipito-temporal-cerebellar cluster, with the highest peaks in the right cerebellum and left occipital fusiform gyrus. Smaller clusters comprised the left and right pre- and postcentral gyri, the left pSTG and classical auditory areas (planum temporale, Heschl's gyrus), the right STG, and the left cerebellum (lobules VIIIa/

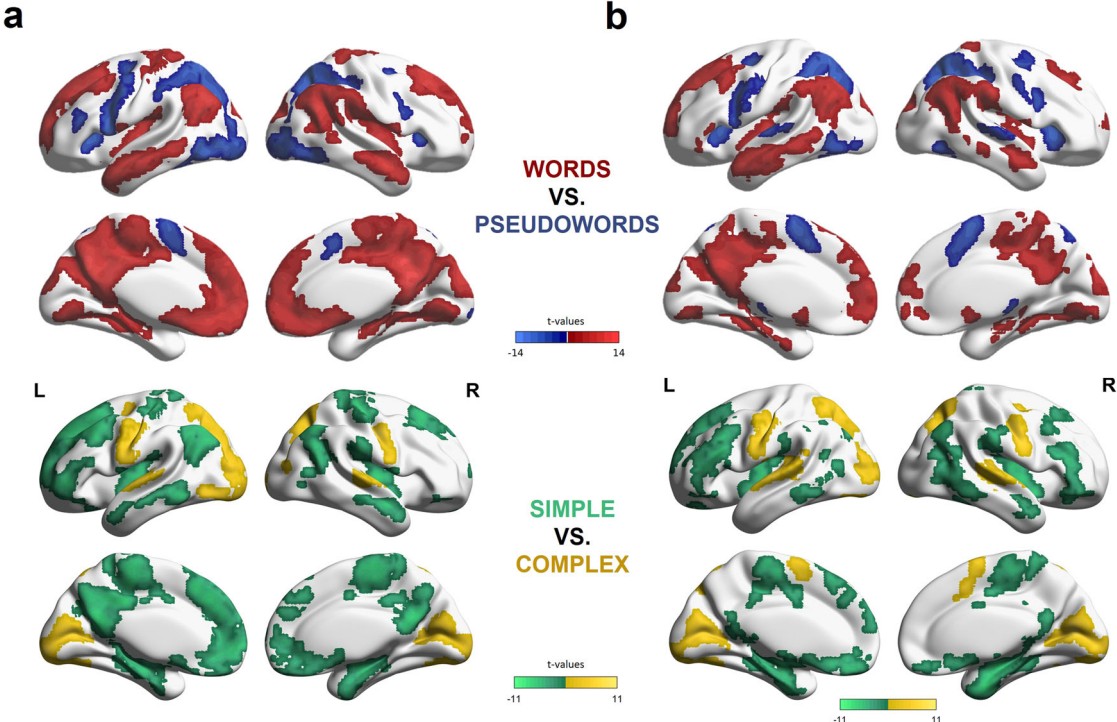

**Fig. 2 Functional activation for the direct contrasts between words and pseudowords and simple and complex stimuli in typical readers and adults with dyslexia. a** Univariate analyses in typical readers revealed large differences in functional activation for the direct contrasts between words (red) vs. pseudowords (blue) and simple (green) vs. complex (yellow) stimuli. **b** The same analyses in the dyslexia group showed largely similar results, suggesting a large overlap in activation patterns between the two groups. All functional activation maps are thresholded at $p < 0.001$ (voxel-level) and $p < 0.05$ (cluster-wise FWE corrected).

VIIb). In contrast, simple stimuli showed consistent recruitment of the left posterior TPC, the bilateral dorsomedial frontal cortex and the MTGs, as well as the insula. Interestingly, the patterns of activation for complex stimuli resembled activation patterns for pseudowords, suggesting higher cognitive demands for pseudoword processing, like the processing of longer and more complex stimuli.

During word and pseudoword reading, univariate analyses revealed hypoactivation in adults with dyslexia in distributed regions (Fig. 3): (1) left vOTC, (2) left supramarginal gyrus (SMG), (3) right vOTC, (4) right cerebellum, and (5) bilateral lingual gyri (Supplementary Table 6). These patterns seemed to be largely driven by pseudoword processing. Word processing only showed group differences in the left vOTC and left SMG, suggesting a pseudoword-specific deficit of the other regions. The direct comparison between words and pseudowords revealed that typical readers had significantly more activation during pseudoword relative to word processing in the right cerebellum and left vOTC.

Since univariate analyses are insensitive to differences in fine-grained activity patterns, we also performed a complementary multivariate pattern analysis (MVPA). Activity patterns in five regions enabled above-chance decoding of group status (adults with dyslexia vs. control group), irrespective of stimulus type (word or pseudoword): the left and right cerebellum, the bilateral precuneus, the left vOTC, and the bilateral paracingulate and anterior cingulate gyri (Fig. 3). For pseudoword reading alone, above-chance decoding was achieved in the left SMG. For word reading alone, activation in the left insula/frontal operculum allowed for decoding between groups. Therefore, in addition to the regions uncovered by the standard univariate analysis, MVPA results suggest that individuals with dyslexia differ from controls also regarding activation in the left insula/frontal operculum

(especially during word reading), bilateral anterior cingulate, right precuneus and left cerebellum.

We wanted to further investigate the observed hypoactivation as revealed by the univariate analysis and therefore chose the five hypoactive regions as regions of interest (ROI) for additional analyses. Looking at the individual activation magnitude within these ROIs for the four conditions, we found that activation was largely consistent within groups (Fig. 4). These individual parameter estimates revealed that the left vOTC, the right cerebellum and the right vOTC showed a linear increase in activation with increased difficulty (from simple to complex and from word to pseudoword stimuli) in the control group. In contrast, the left SMG and the bilateral lingual gyri showed stronger activation for pseudowords and complex stimuli, respectively, in the control group. Parameter estimations from the five ROIs in the dyslexia group revealed practically no condition-specific activation in the left and right vOTC, and the right cerebellum. Furthermore, the observed increase in activation for pseudowords in the left SMG of typical readers and the complexity-specific activation in the bilateral lingual gyri were absent in adults with dyslexia.

**Functional task-related connectivity.** Psychophysiological interactions (PPIs) revealed strong group differences in functional connectivity with adults with dyslexia showing less functional coupling to and from all seed regions as chosen from the univariate contrasts (left vOTC, left SMG, right vOTC and right cerebellum) (Fig. 5, Supplementary Table 7). Functional coupling between the left vOTC and numerous other brain regions was weaker in the dyslexia group during pseudoword reading. These areas included, among others, the bilateral cuneus/superior lateral occipital regions, the right TPC and the right vOTC (Fig. 5a). Moreover, the left SMG exhibited significantly weaker functional

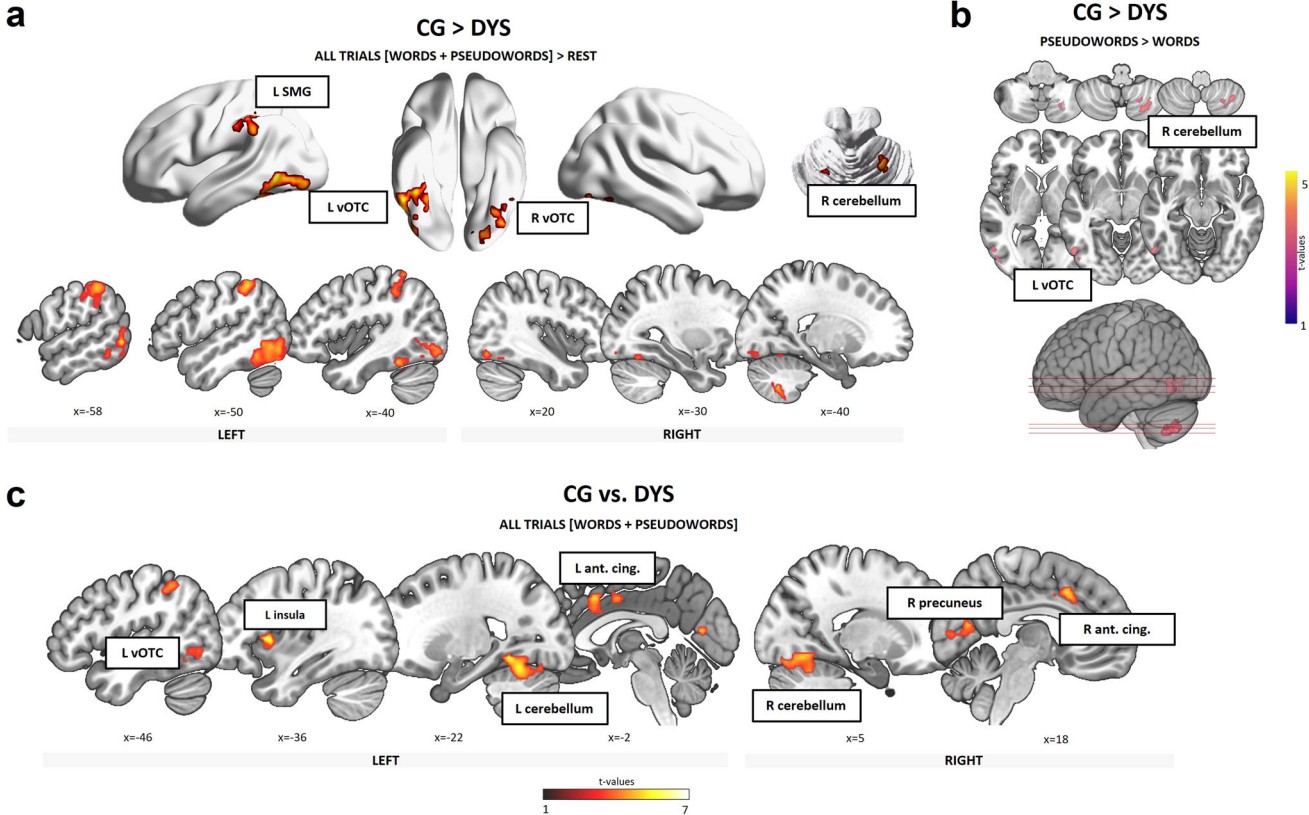

**Fig. 3 Significant differences in functional recruitment of reading areas revealed by univariate and multivariate analyses. a** Univariate analyses show hypoactivation in individuals with dyslexia (DYS) relative to the control group (CG) in several regions in- and outside the classical reading network, such as the left SMG, left and right vOTC and right cerebellum. **b** Univariate analyses for the direct comparison of words and pseudowords show substantial hypoactivation in the right cerebellum and the left vOTC during pseudoword reading specifically. **c** Multivariate analyses revealed additional brain regions showing differences in recruitment between groups. These comprised the left insula, the bilateral anterior cingulate, the right precuneus and the left cerebellum (extending to the left vOTC) (all analyses: $p < 0.001$ voxel-level, $p < 0.05$ cluster-wise FWE corrected).

coupling during pseudoword reading with a large bilateral cluster extending from the occipital poles to the superior lateral occipital cortices and precuneus. In contrast, during word reading, the left SMG showed weaker functional connections in dyslexia with the left vOTC, and the bilateral cerebellum (Fig. 5b). Seeding in the right vOTC, we found weaker functional coupling during pseudoword reading with left pre-/postcentral gyri and bilateral vOTC-adjacent regions (Fig. 5c). Connectivity between the right vOTC and the left precuneus as well as the right supplementary motor area was also weaker during word reading. The strongest differences in functional connectivity between groups were observed for the right cerebellum (Fig. 5d). This region showed less coupling with distributed areas across the bilateral superior frontal, postcentral and superior parietal cortices, as well as the OTC during pseudoword reading, and bilateral supplementary motor area and pre-/postcentral gyri during word reading.

The vOTC regions were the only seed regions for which adults with dyslexia showed increased coupling relative to normal readers (see plots in Fig. 5a, c): during word as compared to pseudoword reading, adults with dyslexia showed stronger functional coupling between the right vOTC and the right cuneus/supracalcarine cortex. When we disentangled this interaction, we found that adults with dyslexia showed a selective disruption (weaker connectivity) of the left and right vOTC and the bilateral precuneus during pseudoword reading, while they engaged this connection similarly to typical readers during word reading.

**Effective connectivity within reading networks**. Based on the literature, reading is thought to rely primarily on three brain regions: the left IFG, the left TPC and the left vOTC. In the control group, our effective connectivity model computed with dynamic causal modelling (DCM) including these three regions showed that all trials drove all three areas. Significant intrinsic connections were found from the left TPC to the left IFG and the left vOTC (both excitatory), and from the left vOTC to the left TPC and the left IFG (inhibitory). Words and pseudowords significantly facilitated coupling from the left IFG to the left vOTC. Moreover, pseudowords positively modulated the connection from the left vOTC to the left TPC. In contrast, word reading turned the facilitatory intrinsic connection from the left TPC to the left vOTC and to the left IFG into inhibition (Fig. 6a – CG; Supplementary Table 8).

In the dyslexia group, all trials drove all three areas and all three areas were reciprocally connected with each other. While the left TPC was inhibited by the left vOTC and the left IFG, the left TPC exhibited an excitatory influence on both regions. The left IFG and the left vOTC inhibited each other. Words modulated all connections except for the vOTC-IFG and TPC-IFG connections. With respect to the direction, words turned the facilitatory connection from the left TPC to the left vOTC into inhibition but reversed the inhibitory influence from the left IFG to the left vOTC and TPC and from the left vOTC to TPC. Pseudowords, in contrast, selectively modulated the dorsal pathway from the left vOTC to left TPC and from the left TPC

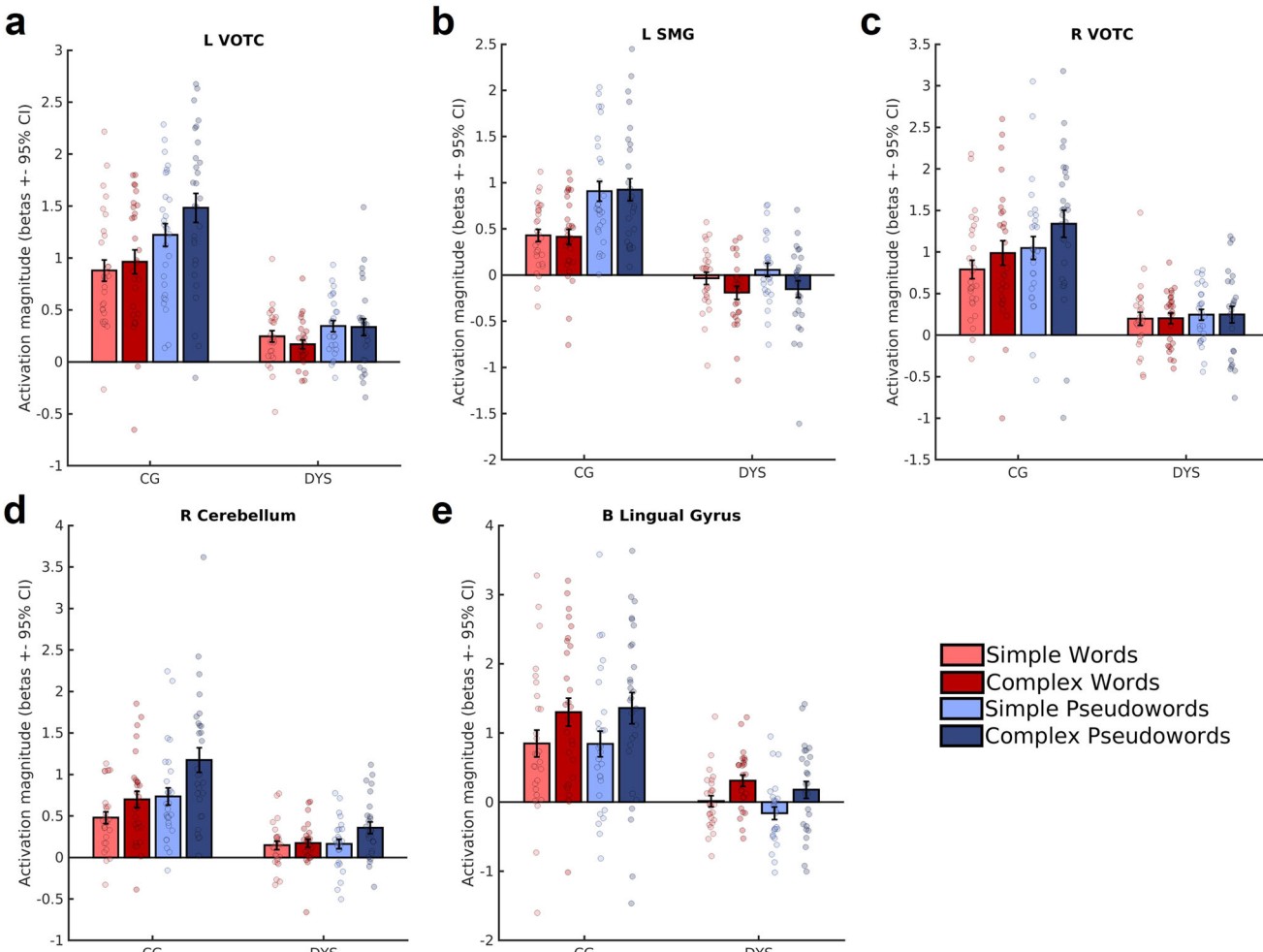

**Fig. 4 Individual activation profiles in hypoactive brain areas during the reading of simple and complex words and pseudowords.** Parameter estimates are based on the respective rest contrasts (simple words > rest, complex words > rest, simple pseudowords > rest, complex pseudowords > rest) in the five hypoactive areas, left vOTC (**a**), left SMG (**b**), right vOTC (**c**), right cerebellum (**d**) and bilateral lingual gyri (**e**). We found high consistency in activation within groups, suggesting that the reported hypoactivation is consistent across subjects in the dyslexia group (DYS; $n = 26$). Additionally, the control group (CG; $n = 27$) showed an increase in activation of the left and right vOTC for increasing difficulty, and a pseudoword-specific recruitment of the left SMG. Also, the bilateral lingual gyri showed increased activation for complex stimuli in the CG. The only activity patterns observable in both groups were that complex pseudowords led to increased activity in the right cerebellum. Bar plots show the mean activation magnitude for each condition and error bars represent one standard error of the mean.

to left IFG, as well as the reverse IFG-to-TPC connection (Fig. 6a – DYS; Supplementary Table 9). For pseudowords, all modulations were facilitatory.

Directly comparing the two groups in a separate DCM analysis revealed three highly significant differences: (a) intrinsic connectivity from the left TPC to the left IFG was significantly stronger in controls; (b) the left TPC showed stronger self-inhibition in the dyslexia group; and (c) words decreased the inhibition of the left vOTC on the left TPC more strongly in adults with dyslexia than controls (Fig. 6a - CG vs. DYS; Supplementary Table 10).

Moving beyond classical reading regions, we performed an additional DCM analysis exploring effective connectivity in the extended (hypoactive) reading network identified in our univariate analysis (Fig. 6b). In the control group, all trials drove all four regions. The right vOTC received facilitatory intrinsic connectivity from the left SMG and the left vOTC. The right cerebellum inhibited all other regions. During word reading, the inhibitory intrinsic connectivity from the right cerebellum to the right vOTC turned into facilitation. Finally, pseudowords

increased the facilitatory connectivity from the left SMG to the right vOTC (Fig. 6b - CG; Supplementary Table 11).

In the dyslexia group, only the left and right vOTC received driving input from all stimuli. Several intrinsic connections were modulated by pseudoword and word reading, with pseudowords strengthening the left vOTC to left SMG connection and turning the right cerebellum to right vOTC inhibition into excitation. Apart from that, connections to and from the left vOTC were positively modulated by both stimulus types, especially the left vOTC- right vOTC interaction. Only right vOTC to left SMG coupling was weakened during word and pseudoword reading (Fig. 6b - DYS; Supplementary Table 12).

A direct comparison of the two groups yielded the following results: (a) adults with dyslexia showed stronger intrinsic coupling from the left vOTC to the right cerebellum, whereas typical readers showed stronger intrinsic connectivity from the left vOTC to the right vOTC, and from the right vOTC to the right cerebellum; (b) words modulated effective connectivity from the left vOTC to the right cerebellum more strongly in dyslexia,

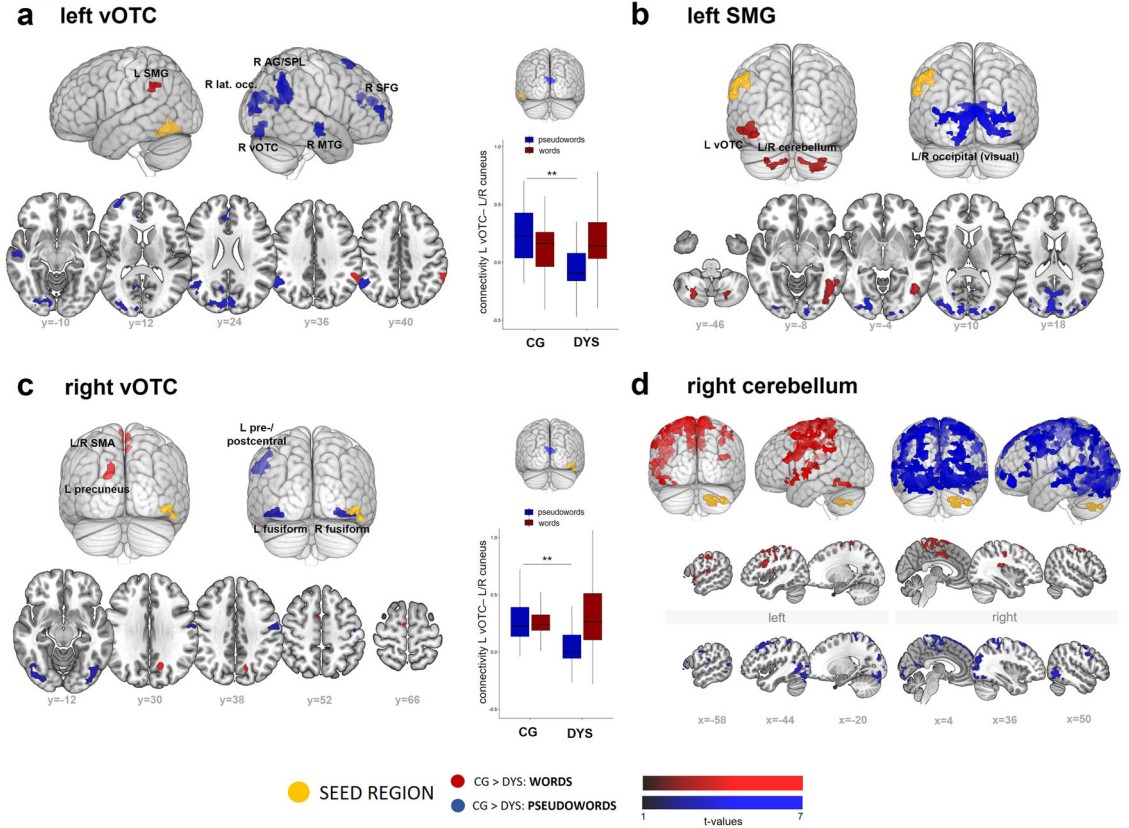

**Fig. 5 Differences in task-specific functional coupling between the control group and the dyslexia group.** We display PPI results for seeds from the univariate contrast results: (**a**) left vOTC, (**b**) left SMG, (**c**) right vOTC, (**d**) right cerebellum ($p < 0.001$ voxel-level, $p < 0.05$ cluster-wise FWE corrected). The plots depict the respective seed region (yellow) and areas which revealed significantly more connectivity with the seed region in typical readers relative to individuals with dyslexia during word (red) and pseudoword (blue) reading. We found disruptions of functional connectivity between left SMG and left vOTC, as well as between left SMG and bilateral cerebellar and visual areas. The strongest effects were found for the right cerebellum (**d**), where adults with dyslexia (DYS, $n = 26$) showed much lower functional connectivity during word and pseudoword reading to large clusters spread across the two hemispheres. For connectivity between the left vOTC and left/right cuneus, we found stronger coupling for individuals with dyslexia than controls (CG, $n = 27$) (see box plots in (**a**) and (**c**), displaying the respective connectivity strength). This significant effect, however, did not stem from stronger connectivity between these regions in adults with dyslexia, but from the difference in recruitment for word and pseudoword reading. Box plots in (**a**/**c**) show median connectivity for each condition and error bars represent the maximum and minimum values.

and from the right vOTC to the right cerebellum in typical readers; (c) pseudowords modulated effective connectivity from the left vOTC to the left SMG, and from the left vOTC to the right cerebellum more strongly in dyslexia, while in typical readers, pseudowords showed a stronger modulation of the connectivity from the right vOTC to the left SMG; (d) self-inhibition of the right vOTC was higher in the dyslexia group, whereas self-inhibition of the right cerebellum was higher in the control group (Fig. 6b - CG vs. DYS; Supplementary Table 13).

**Brain-behaviour links.** In terms of functional activation, we found strong correlations between activation in the hypoactive brain areas (left SMG, left vOTC, right vOTC, right cerebellum and bilateral lingual gyri) and reading performance (in and outside the scanner) across groups. Significant correlations are reported in Table 2, whereas all results including non-significant results are presented in Supplementary Table 14. These correlations suggest that the left SMG and the cerebellum show stronger pseudoword-specific recruitment, whereas the left and right vOTC and bilateral lingual gyri show slightly stronger links to word reading. Please note that these strong correlations were not significant within groups. Correlational plots are also provided in Supplementary Figs. 5–11.

In the next step, we extracted PPI functional connectivity values for every single participant and correlated them with in- and out-of-scanner reading performance across groups. Since we found less functional coupling in numerous areas in the dyslexia group in our PPI analysis (see Fig. 5), we decided to limit the correlations to functional connections between areas that we designated as classical and extend reading network nodes in our DCM analysis (see Fig. 7). These comprised nine connections of interest starting always with the respective seed region: (1) left SMG - left vOTC, (2) left vOTC - left SMG, (3) left SMG - right cerebellum, (4) right cerebellum - left vOTC, (5) left vOTC - right vOTC, (6) right cerebellum - left temporal cortex (the cluster as found in the PPI comprised large portions of left posterior STG and posterior MTG), (7) right cerebellum - left IFG, (8) right cerebellum - left vOTC and (9) right cerebellum - right vOTC. Bayesian correlations with a $BF_{10} > 7$ are reported in Table 3. The whole range of results, including non-significant findings, is provided in Supplementary Table 15.

Connectivity between left SMG and left vOTC was strongly tied to word reading performance ($\tau = 0.347$, $BF_{10} = 130$; $\tau = 0.355$, $BF_{10} = 171.88$) during our behavioural assessment, both when the left SMG and the left vOTC were taken as seeds in the PPI. Moreover, word reading performance outside the scanner was tightly linked to connectivity between the left SMG

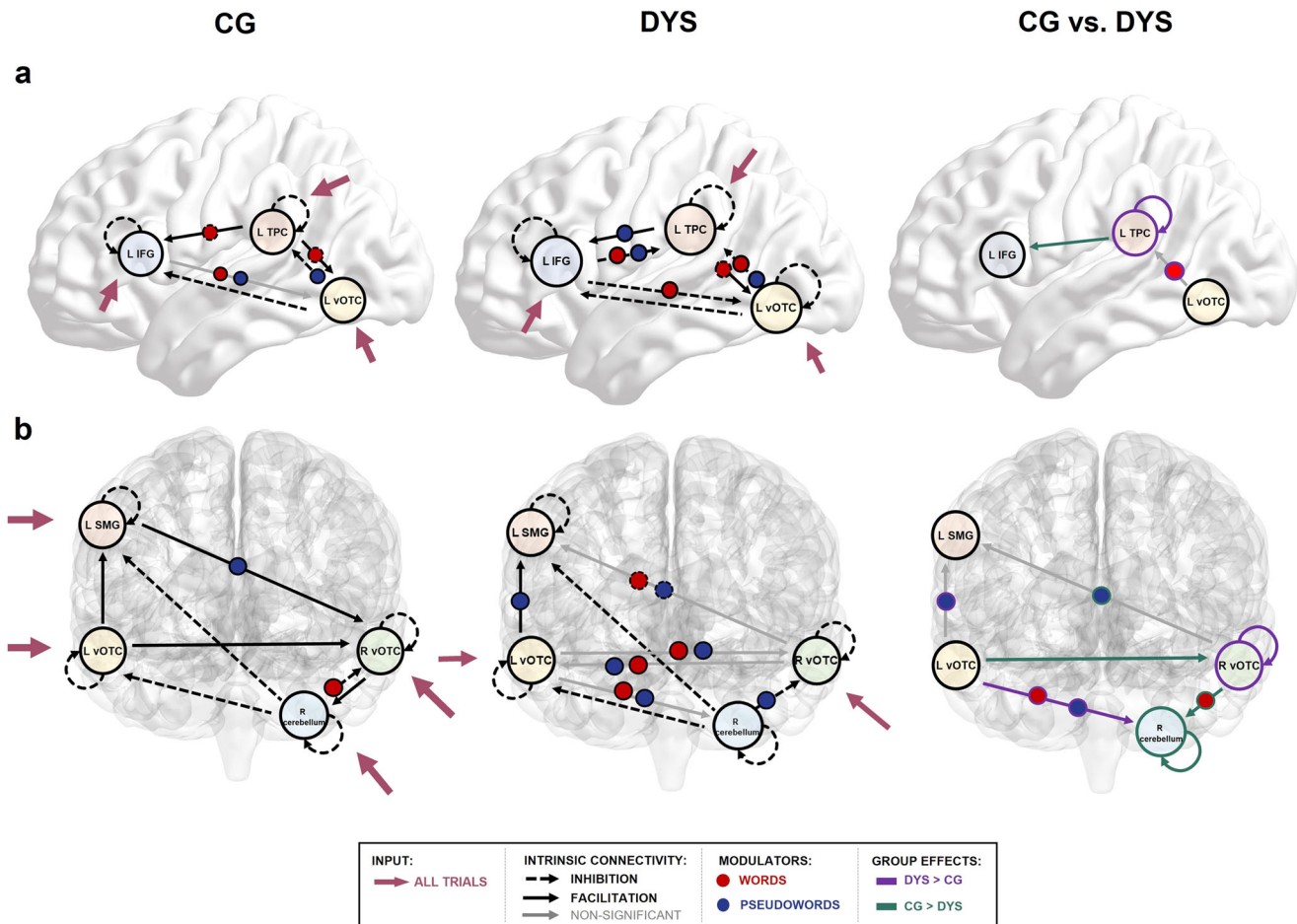

**Fig. 6 Differences in effective connectivity between adults with and without dyslexia in the classical and the extended reading network.** We display the best DCM model for each group (posterior probability >95), as well as a comparison of DCMs across groups. **a** We report differences in connectivity within the classical reading network (including left IFG, left TPC and left vOTC). In the control group (CG), words decreased the connectivity via the dorsal route (left TPC to other areas), whereas pseudowords partially relied upon this route. In adults with dyslexia (DYS), in contrast, both words and pseudowords recruited connectivity to and from the left TPC. A direct comparison between both DCMs showed significantly stronger recruitment of the dorsal route during word reading and a stronger self-inhibition of the left TPC in dyslexia, and stronger left TPC to IFG coupling in typical readers. **b** DCMs of the extended reading network (including left SMG, left vOTC, right vOTC and right cerebellum) revealed almost no modulation by stimulus type and strong control of the left vOTC over the left SMG and the right vOTC in typical readers. In adults with dyslexia, both words and pseudowords differentially recruited the various pathways between regions of the extended reading network. Differences between the groups were evident in neural interactions with the right cerebellum and in reliance upon the left SMG for pseudoword reading.

and the right cerebellum ($\tau = 0.304$, $BF_{10} = 27.53$), as well as the right cerebellum and left vOTC ($\tau = 0.298$, $BF_{10} = 22.74$). Connectivity from the right cerebellum to three other regions (left temporal: $\tau = 0.270$, $BF_{10} = 8.94$; left IFG: $\tau = -0.305$, $BF_{10} = 25.95$; right vOTC: $\tau = 0.266$, $BF_{10} = 8.47$), in contrast, correlated strongly with pseudoword reading performance (in and outside the scanner). Higher connectivity values were always associated with better performance. We thus suggest that word reading performance was linked to an overall stronger functional interaction between left SMG and left vOTC. Better pseudoword reading performance, in contrast, was linked to stronger connectivity between the right cerebellum and bilateral reading-related areas (left IFG, left temporal cortex, right vOTC) (see Supplementary Figs. 8, 9). Please note that these findings were only significant when looking at reading ability as a continuum and including both groups, but not within groups and should thus be interpreted with caution.

In a final step, we explored how directed functional coupling, that is, when one region exerts an inhibitory or facilitatory influence on another region, correlated with behavioural

performance in each group. To do so, we computed correlations between reading performance and Hz values as derived from the DCMs. We found strong links for coupling of the left vOTC to the other areas within the classical reading network: lower inhibition (i.e., more positive coupling) of the left vOTC to the left IFG went hand in hand with lower inhibition of its connection to the left TPC (and vice versa) ($\tau = 0.464$, $BF_{10} = 35$). In terms of links to behaviour, we found that lower intrinsic connectivity from the left TPC to the left vOTC was associated with higher simple word reading accuracy ($\tau = -0.378$, $BF_{10} = 9.65$) (Fig. 7a).

Looking at effective connectivity parameters within the classical reading network in the dyslexia group, we found that a higher modulation of left vOTC to left TPC coupling by pseudowords was linked to a lower modulation of left IFG to left TPC coupling by pseudowords ($\tau = -0.385$, $BF_{10} = 9.47$). In other words, pseudoword processing increased the connectivity from the left vOTC to the left TPC, while coupling from the left IFG to the left TPC decreased. Behaviourally, we found that the modulation of words on the left TPC to left vOTC connection

was linked to simple word reading accuracy ($\tau = -0.419$, $BF_{10} = 18.55$). A more negative modulation of that connection by words was linked to better simple word reading performance. In contrast, intrinsic connectivity from the left TPC to the left IFG correlated with speech onsets for words ($\tau = 0.385$,

$BF_{10} = 9.47$). Here, higher connectivity was linked to slower word reading (Fig. 7b).

Within the extended, hypoactive reading network, the control group showed a strong correlation between connectivity from the right cerebellum to the left vOTC, and to the left SMG ($\tau = 0.481$, $BF_{10} = 93.06$). The more negative (inhibitory) the connectivity from the right cerebellum to the left SMG, the more negative the connectivity to the left vOTC, and vice versa. In line with the DCM findings within groups, this confirmed the right cerebellum's role as an inhibitory control area in neurotypical readers. Moreover, the facilitatory influence of the left SMG to the right vOTC was negatively linked to the intrinsic coupling of the left vOTC to the right vOTC ($\tau = -0.419$, $BF_{10} = 22.09$). In this case, more positive coupling from the left vOTC to the right vOTC co-occurred with lower functional coupling from the left SMG to the right vOTC.

In the dyslexia group, we found that left vOTC to left SMG coupling correlated with complex pseudoword reading times ($\tau = 0.348$, $BF_{10} = 8.44$) (a weaker effect was also present for simple pseudowords). That is, faster reading of complex pseudo-words went hand in hand with lower functional coupling from the left vOTC to the left SMG. Regarding word processing, the connectivity from the right cerebellum to the right vOTC correlated with speech onsets for simple words ($\tau = 0.372$, $BF_{10} = 7.54$) (a weaker effect was also present for complex pseudowords). Stronger inhibition (i.e., more negative functional connectivity) exerted by the right cerebellum on the right vOTC was associated with longer speech onsets for simple words (Fig. 7b).

## Discussion

The present study investigated neural network interactions underlying reading deficits in adults with dyslexia. We found significantly reduced functional connectivity between reading nodes and domain-general brain regions in people with dyslexia

**Table 2 Bayesian correlations for functional activation and reading performance.**

| ROI | Performance | $\tau$ | $BF_{10}$ |
|---|---|---|---|
| L vOTC | pseudoword reading | 0.323 | 52.78 |
| | pseudoword reading times (MRI) | −0.273 | 9.75 |
| | pseudoword accuracy (MRI) | 0.304 | 24.11 |
| | word reading | 0.395 | 911.54 |
| L SMG | pseudoword reading | 0.497 | 48686.61 |
| | pseudoword speech onsets (MRI) | −0.309 | 30.1 |
| | pseudoword reading times (MRI) | −0.422 | 2509.20 |
| | pseudoword accuracy (MRI) | 0.334 | 71.32 |
| | word reading | 0.404 | 1334.20 |
| | word speech onsets (MRI) | −0.27 | 8.93 |
| R vOTC | pseudoword reading | 0.309 | 33.53 |
| | pseudoword accuracy (MRI) | 0.272 | 9.34 |
| | word reading | 0.344 | 116.50 |
| R cerebellum | pseudoword accuracy (MRI) | 0.304 | 25.11 |
| | word reading | 0.276 | 11.47 |
| B lingual | pseudoword reading | 0.302 | 26.28 |
| | word reading | 0.347 | 364.66 |
| | word speech onsets (MRI) | −0.262 | 7.20 |

We found correlations for activation within the five hypoactive ROIs from the univariate analyses (CG > DYS [pseudowords > rest], [words > rest]), namely left vOTC, left SMG, right vOTC, right cerebellum and bilateral lingual gyri, with in- and out-of-scanner performance (full sample comprising both groups; n = 53). Out-of-scanner performance comprised word and pseudoword reading, whereas in-scanner performance included pseudoword/word speech onsets, reading times and accuracy (tagged with MRI). All regions were linked to both types of stimuli, with stronger effects on one of the two stimulus types, however. We report Kendall's τ for robust correlations with a $BF_{10} > 7$[71].

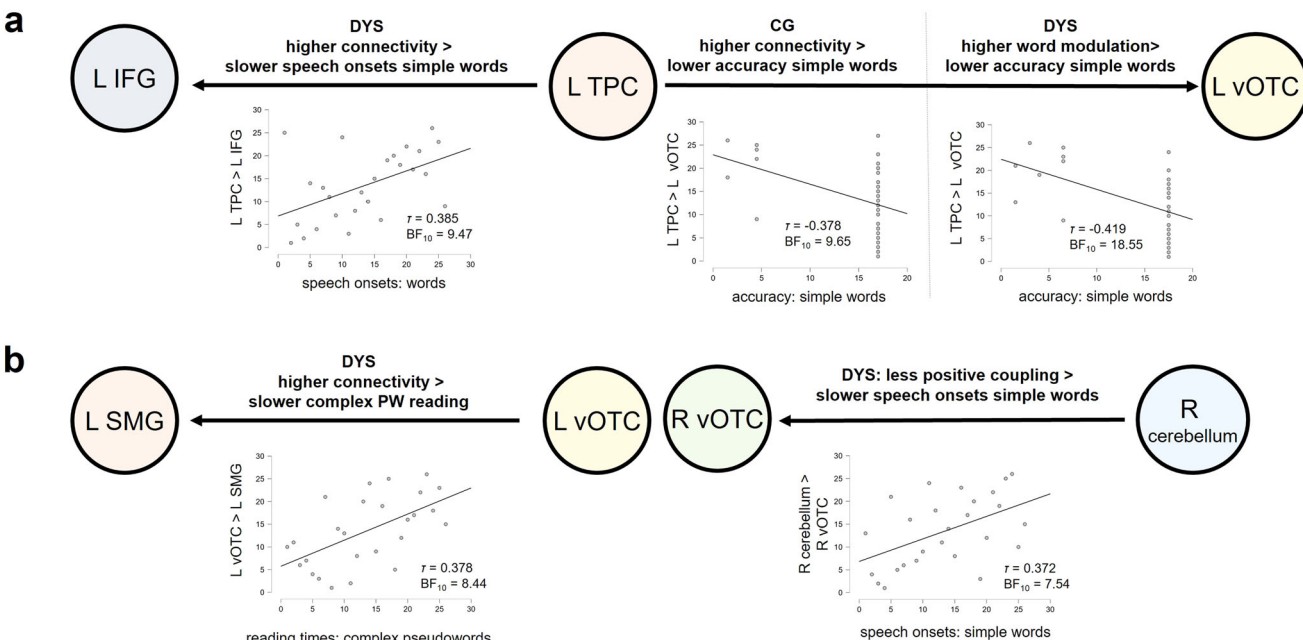

**Fig. 7 Significant correlations between effective connectivity within reading networks and reading performance (n = 53). a** We display Bayesian correlations (BF$_{10}$ > 7) between measures of effective connectivity within the classical reading network and task performance of in-scanner performance in adults with (DYS) and without dyslexia (CG). **b** We display Bayesian correlations (BF$_{10}$ > 7) between measures of effective connectivity within the extended (hypoactive) reading network and in-scanner task performance. Across all correlations, higher connectivity in the dyslexia group was linked to worse reading performance. In other words, individuals with the strongest coupling showed the weakest performance. While left SMG to left vOTC coupling seemed to be crucial for complex pseudoword reading in dyslexia, left TPC to IFG and to left vOTC coupling were more important for word reading.

**Table 3 Bayesian correlations for connections of interest derived from PPI analyses and reading performance.**

| Connections of interest (PPI) | Performance | τ | BF₁₀ |
|---|---|---|---|
| L SMG - L VOTC | Word reading | 0.347 | 130.04 |
| L vOTC - L SMG | Word reading | 0.355 | 171.88 |
| L SMG - R Cerebellum | Word reading | 0.304 | 27.53 |
| R Cerebellum - L vOTC | Word reading | 0.298 | 22.74 |
| R Cerebellum - L Temporal | Pseudoword accuracy (MRI) | 0.270 | 8.94 |
| R Cerebellum - L IFG | Pseudoword reading times (MRI) | −0.305 | 25.95 |
| R Cerebellum - R vOTC | Pseudoword reading | 0.266 | 8.47 |

Out-of-scanner performance comprised word and pseudoword reading, whereas in-scanner performance (tagged with MRI) included pseudoword/word speech onsets, reading times and accuracy (full sample comprising both groups; n = 53). Whereas correlations with word reading comprised only out-of-scanner performance and specifically the left SMG to left vOTC coupling, pseudoword performance was tied to connections originating from the right cerebellum. We report Kendall's τ for robust correlations with a BF₁₀ > 7[71].

compared to neurotypical readers. Additionally, the observed strong decoupling of the right cerebellum with distributed areas in both hemispheres for word and pseudoword reading suggests that aberrant right cerebellar connectivity may be a biomarker for dyslexia[32]. This would support and extend earlier theories on cerebellar dysfunction resulting in the impaired automatization of high-order sensory-motor procedures in dyslexia[33,34]. Accordingly, our effective connectivity analyses showed that stronger inhibitory connectivity from the right cerebellum to the right vOTC was associated with less efficient (slower) simple word reading in adults with dyslexia. Effective connectivity analyses further suggest that neurotypical readers relied primarily upon interactions in the classical reading network. Atypical readers, in contrast, showed an overall increase in the number of interactions both in the core and extended reading network for word and pseudoword reading, and a relatively stronger contribution of the dorsal stream. Overall, this suggests a multifocal account of dyslexia visible in functional activation and underlying functional and effective connectivity between specific brain regions[35].

Behavioural assessments and task performance showed persisting reading difficulties, especially pseudoword-specific weaknesses, in adults with dyslexia. In line with earlier studies, adults with dyslexia performed significantly worse during administered tasks, ranging from word, pseudoword and text reading to phoneme substitution, and spelling[36,37]. We report similar findings for task performance during fMRI, where the dyslexia group had significantly slower speech onsets for simple and complex words and pseudowords, and slower reading times and lower accuracy for pseudowords. This difficulty in pseudoword reading supports the assumption of ongoing indirect/sublexical processing deficits and associated phonological decoding issues persisting into adulthood[38].

Our findings support earlier research on the neurofunctional profiles of word and pseudoword reading and complexity processing. In accordance with an earlier meta-analysis, we found that pseudowords showed stronger recruitment of the left precentral gyrus and IFG area, whereas words engaged classical semantic regions like the left AG more strongly[6]. When directly comparing words > rest and pseudowords > rest, we observed activation in similar regions in both groups. However, when directly contrasting the two conditions, we found that due to the deactivation of various areas during word reading, large differences in functional recruitment arose. Concerning complexity, we found similar networks as revealed by the word–pseudoword comparison, with complex items relying on the engagement of similar regions as pseudowords.

We confirm that dyslexia is characterized by marked hypoactivation in classical reading areas. Our univariate findings support a difference in left-hemispheric recruitment for words and pseudowords. The left-hemispheric hypoactivation in the left

TPC and OTC corroborates earlier findings of meta-analyses on functional activation in dyslexia[14,15,39]. Interestingly, however, our results of the right cerebellum stand in contrast to recent work linking it to compensation in shallow orthographies[40] and presenting it as a functional compensatory marker for dyslexia in children[41]. In the present study, we did not detect compensatory activation as suggested by other studies[15,31,41] but the differences in recruitment in the areas revealed by MVPA could be linked to attempts for compensation. MVPA identified fine-grained differences in functional activation patterns in domain-general regions across the two hemispheres. These regions included the left insula, the bilateral anterior cingulate and the right precuneus. Whereas a dysfunctional insula was suggested as a marker for dyslexia already in early studies[42] and is supported by a recent meta-analysis[43], anterior cingulate activation is usually linked to increased attention and effort[44] and negatively linked to reading performance[45]. In earlier studies, the right precuneus was less activated in adults with dyslexia when compared to controls during working-memory-related tasks[46,47]. Activation differences in the precuneus and cingulate gyri could thus be interpreted as deficits in fully engaging domain-general processing regions that support reading processes vital for reading skills in dyslexia[48]. Last, we also found differences in activity patterns of the left cerebellum. Activation in the cerebellum has been implicated in higher cognitive functions and executive control, and has recently been linked to semantic processing[23,49].

Typical readers showed differential and consistent recruitment of reading-related regions depending on the specific task. In typical readers, the bilateral vOTC and the right cerebellum showed a linear increase in activation magnitude with increasing difficulty/complexity – from words to pseudowords and simple to complex stimuli. Additionally, typical readers showed a stronger recruitment of the left SMG during pseudoword stimuli and of the bilateral lingual gyri for complex stimuli. The anterior dorsal SMG has been linked to pseudoword reading in an earlier study that explored the SMG subdivisions' recruitment during cognitive tasks[50], which might well explain the consistent recruitment of this region during the two pseudoword conditions. The lingual gyri, in contrast, have been suggested to show higher activation due to increasing word length, that is, increasing demands on local feature processing have probably led to heightened activation in the lingual gyri[51]. None of these effects were observable in the dyslexia group, where activation in these hypoactive regions was largely consistent across subjects, but generally very low. Based on our analyses, we confirm consistent hypoactivation with no large differences between individuals in our dyslexia group and suggest pseudoword-specific recruitment of the left SMG in typical readers.

Overall, typical readers seem to possess more widespread and consistent functional connectivity between brain regions. We

found that left-hemispheric reading regions, such as the left SMG and vOTC, showed reduced functional interactions in adults with dyslexia. Interestingly, we did not detect any regions displaying higher functional coupling in adults with dyslexia. This stands in contrast to an earlier study that found primarily a disruption of left TPC/OTC to left IFG coupling[52], and higher functional connectivity from reading-related areas to default mode network regions in dyslexia[25]. We found the inverse pattern during overt pseudoword reading, with a clear disruption of functional coupling between the bilateral vOTC and the bilateral precuneus. The precuneus is known as the inhibitory control hub at the intersection between several brain networks[53]. Since pseudoword reading requires increased cognitive control due to high phonological decoding demands, a disruption of this area could be linked to reading difficulties, such as observable in individuals with dyslexia. For pseudoword reading, connectivity between the left vOTC and the right hemisphere also seemed largely deficient, which was supported by our DCM findings. The left SMG showed differences in functional connectivity with the bilateral cerebellum for word processing, probably tied to a semantic and/or phonological involvement of these regions[23]. The areas less connected with the left SMG during pseudoword reading were mainly visual/occipital areas. The right vOTC interacted less with domain-general regions for word reading, suggesting a basis for less automatic processing in dyslexia. Moreover, the right vOTC also showed lower connectivity with visual (fusiform) and motor regions during pseudoword reading. These findings indicate that generative spelling-sound knowledge depends more on connectivity to the right hemisphere and bilateral visual processing regions, while words rely more on domain-general and typical reading regions, as well as the cerebellum. Last, the right cerebellum revealed altered effective connectivity in adults with dyslexia, reflected in extensive differences in functional coupling with numerous regions spread across both hemispheres. This emphasizes the significant role of the right cerebellum, even if, as indicated by our DCM findings, it might mainly exert an inhibitory role during reading processes due to its connection to the posterior parietal cortex, frontal eye fields and dorsolateral prefrontal cortex during visual processes[54].

Correlations between functional activation, functional connectivity and behavioural performance showed strong effects when exploring the whole range of reading-related abilities. Within groups, we did not find any strong correlations between activation and connectivity with reading performance. We believe that this was due to small variability in reading performance (e.g., ceiling effects for word performance in the scanner in controls) and activation in the dyslexia group (see Fig. 4). However, when we shifted away from our group status and correlated individual reading performance across groups with functional activation, we found strong links between activation in left SMG and right cerebellum and pseudoword reading. Left vOTC, right vOTC and bilateral lingual gyri showed stronger correlations with word reading. In terms of functional connectivity, left vOTC and left SMG connectivity was strongly tied to word reading. Interestingly, connectivity between the right cerebellum and left temporal, left inferior frontal and right vOTC regions was tied to pseudoword performance in the scanner only. Connectivity between the right cerebellum and left SMG and left vOTC, on the other hand, was tied to word reading performance (see schematic illustration of findings in Supplementary Fig. 12).

Across groups, effective connectivity analyses revealed that the left vOTC and the right cerebellum inhibited other reading regions. DCMs of the classical and extended reading network found inhibitory influences of these two areas in both groups, except for the left SMG. This region was always positively modulated by the left vOTC. As the cerebellum is believed to play a role in inhibition and executive functioning[55], specifically for regulating attentional orientation[56], its function as an inhibitory control region during reading seems plausible. A disconnection between the right cerebellum and supratentorial language regions has been found in some disorders (e.g., autism[57]), hinting at its role in learning processes relying upon attention. Directly comparing the two groups, we found that controls showed no left-hemispheric interaction with the right cerebellum, in contrast to adults with dyslexia. The right vOTC showed positive coupling with the right cerebellum in controls, supporting our PPI findings showing a severe disruption of coupling with the right cerebellum in dyslexia.

Group comparisons of effective connectivity provide important insight into the recruitment of the dorsal reading route in dyslexia (see Supplementary Fig. 13). We found that neurotypical readers relied less on the dorsal route during word processing, while adults with dyslexia engaged the dorsal route during both word and pseudoword reading. The dorsal route is thought to be mainly involved in phonological decoding (sound-to-letter-mapping) and should thus mainly be engaged during pseudoword reading[4,58]. However, we report differential recruitment of subregions of the TPC, namely the left pSTG and the left SMG (see Fig. 7). Generally, recruitment of the dorsal route for word reading might suggest that words are still not fully automatized (note that the effect might be different for simple and complex words) in dyslexia and require phonological decoding. However, since we found rather an engagement of the left pSTG than the left SMG for word reading in dyslexia, it might not be phonological decoding per se that is still required, but access to auditory and phonological representations that is more important in individuals with dyslexia to read typical words. Interestingly, the direct group comparison also showed increased coupling from the left vOTC to the left SMG during pseudoword reading in dyslexia, supporting our assumption that it is rather this subregion that is involved in phonological decoding. The stronger reliance upon this connection in adults with dyslexia as compared to typical readers most likely hints towards a larger reliance and a higher demand of this area for pseudoword processing. Last, we found stronger intrinsic connectivity from left TPC to left IFG in neurotypical readers which likely reflects a fast and efficient feedforward connection for phonological decoding to speech production and supports earlier theories of disrupted temporo-frontal connectivity in dyslexia[52].

Overall, neurotypical readers showed strong coupling between nodes of the extended reading network with few task-specific modulations, whereas individuals with dyslexia showed less stable connectivity. This is particularly true for the recruitment of the bilateral vOTC within the extended reading network: controls relied upon unidirectional left-to-right coupling and showed no modulations by stimulus type. In dyslexia, in contrast, we observed differential bidirectional coupling between the left and right vOTC, which we interpret as an attempt to integrate the right hemisphere to support reading processing. Generally, communication within the network was rather straightforward in the control group: left vOTC actively interacted with left SMG and right vOTC during reading and was inhibited by the right cerebellum. In dyslexia, the opposite was the case: the left vOTC attempted to interact with the right vOTC and the right cerebellum, while left SMG and left vOTC were modulated by the right vOTC, and at the same time inhibited by the right cerebellum. We believe that this non-straightforward functional coupling largely reflects compensatory attempts due to less efficient processing and reliance upon right-hemispheric regions. Taken together, this supports evidence on differential recruitment of reading circuits in dyslexia[26] and emphasizes the key role of the bilateral vOTC for reading.

The observed links between neural and behavioural findings show how connectivity can impact reading performance. In both the control and dyslexia groups, we found that less coupling from left TPC to left vOTC was linked to higher word reading accuracy, especially for simple words. This suggests that less interaction between the left TPC and left vOTC led to increased reading performance for words (see also Fig. 7). Given that the dorsal stream is implicated in decoding, it is no surprise that less engagement of this stream could lead to better word reading performance. It is interesting, however, that this was also the case for the dyslexia group, in which both words and pseudowords seemed to be processed via a dorsal mechanism. Most likely, the dorsal route was additionally recruited for words due to a higher need for access to auditory and phonological representations, but the stronger the decoupling, the more the ventral route was engaged, which was reflected in higher accuracy. In the dyslexia group, we found additional links to behaviour for regions within and outside the classical reading network. The fact that lower intrinsic coupling from the left TPC to the left IFG was associated with faster reading processing may reflect that those individuals with better performance required fewer interactions between these areas. Since adults with dyslexia seemed to rely upon the dorsal route for word processing as well, it is reasonable to assume that the TPC to IFG connection plays a role in word reading as well. Moreover, less influence of the left vOTC on the left SMG was linked to faster complex pseudoword reading. Since this interaction was only significant in adults with dyslexia, it is likely that the positive correlation between behaviour and decoupling reflects a pattern that resembles the healthy network. In other words, stronger facilitatory influences from the left vOTC to the left SMG may rather reflect aberrant connectivity. Last, we found that less inhibition of the right vOTC by the right cerebellum was linked to faster speech onset for words. Although this intrinsic connection was modulated by pseudowords in our DCM, the connection between the two is implicated in word reading in our control group. Thus, those adults with dyslexia who show less inhibitory drive from the right cerebellum to the right vOTC might be those who resemble more the typical network, which is reflected in the link to faster speech onsets.

Based on our results, we characterize dyslexia as a network disorder by providing insight into task-specific patterns of hypoactivity and hypoconnectivity comprising areas within and outside the classical reading networks. One limitation of the present study is the difficulty of accounting for differences in reading training and compensation over the years, with each individual with dyslexia following their own trajectory of reading development and progress. The observed results might thus not only stem from general differences between typical and atypical readers, as found in children with dyslexia and typically reading peers, but also from compensatory mechanisms and processes influenced by extensive reading instruction, training and compensation strategies that developed over decades. While the sample size is rather small, we did our best to create a homogenous dyslexia group (e.g., by making sure that no subjects had attention or arithmetic deficits, met our stringent criteria for dyslexia status) and used standardized and manifold measures to assess reading abilities. We believe that the findings of the present study can guide future neurostimulation studies, which, for instance, could aim to tackle the observed hypoactivation in the right cerebellum or the left SMG with facilitatory stimulation protocols.

## Methods

**Subjects**. The sample included neurotypical adult readers ($N = 28$; $M_{age} = 28 \pm 5$ years; 18–40 years; 13 females) and adults with dyslexia ($N = 26$; $M_{age} = 26.5 \pm 6$ years; 18–40 years; 17 females). Subjects in the dyslexia group had an official diagnosis and a history of reading and spelling problems and/or scored 1.5 standard deviations below the mean of the control group in >50% of administered reading and spelling tests. None of our participants had any other neurological, neurodegenerative, language and learning disability. All participants were German native speakers residing within Germany and paid for participation. Prior to participation, written informed consent was obtained from each subject. The study was performed according to the guidelines of the Declaration of Helsinki and approved by the local ethics committee of the University of Leipzig.

**Experimental procedure and behavioural assessment**. The overall study comprised a 3-hour behavioural testing session including reading, spelling, and general cognitive testing, and two sessions that combined fMRI and transcranial magnetic stimulation. Please note that to answer the present questions, only data from the ineffective sham (placebo) stimulation session entered the here reported analyses.

During the behavioural session, we did not only assess reading and spelling, but also ruled out that subjects had low nonverbal intelligence and comorbidities that could impact our results. The first subtest of the Culture Fair Test[59] was administered to test participants' nonverbal intelligence. For all tests, points were added up to calculate raw scores which were then converted into standardized scores. Since no subject had a nonverbal IQ below 85, nobody had to be excluded from further participation. Furthermore, we administered two tests to rule out arithmetic and attention deficits: (1) the Eggenberger Rechentest[60], a standardised arithmetic test consisting of calculations, conversions and applied mathematics; (2) the Continuous Attention Performance Test[61], measuring selective and continuous attention.

Verbal working memory was assessed with digit span forward and digit span backward[62]. Subjects had two attempts to repeat the same number of digits. If both were incorrect, the test was terminated. A total of 14 (raw) points could be achieved. Additionally, nonword span was assessed with the Mottier test[63]. The test was terminated when subjects could repeat <50% of the nonword syllables.

In terms of reading, we assessed text, word and pseudoword reading. Silent text reading was assessed[64], providing speed (number of words read), accuracy (ratio of filled gaps and correct items) and comprehension scores (number of correctly inserted words). Participants had six minutes to read as far as they could and fill in missing gaps in the text using one of three options. Word and pseudoword reading was assessed in a speeded reading paradigm[65]. Subjects had to correctly read aloud as many words/pseudowords on a list as possible in one minute. The difficulty increased with each second column. One raw score was computed for speed and accuracy based on the number of correctly read words/pseudowords in one minute. Spelling was assessed by a Rechtschreibungstest[66], in which participants were asked to fill in 68 missing words of a text, which the examiner read out loud.

Phonemic awareness, lexical access and retrieval of phonological representations were tested by applying and recording a spoonerism task (German adaption)[67], where participants were asked to interchange the initial sounds of the first names and surnames of 12 well-known German personalities of cartoon figures. Response time was measured from the offset of the stimulus to the offset of the response. Finally, to assess reading fluency, we used a rapid automatized naming task with letters, colours and numbers[68]. Naming time was measured and a raw score for items read per second was calculated.

**Statistics and reproducibility**. Statistical analyses were performed with JASP[69] and R 4.2.1[70]. To explore behavioural differences between groups, we compared mean scores across the cognitive, reading and spelling sessions using Mann-Whitney-$U$ tests given that several assessed variables were not normally distributed (test statistics are presented in the corresponding table). To investigate in-scanner performance differences, we ran three linear mixed models using the lmer function in R. Speech onsets, reading times and reading accuracy were chosen as dependent variables. Fixed effects were group (control vs. dyslexia), stimulus type (word vs. pseudoword) and complexity (simple vs. complex). As we expected large individual variability in the dyslexia group and ceiling effects for word stimuli, we tested models with random intercepts for subject and trial.

To investigate brain-behaviour relations, we explored correlations between performance and functional activation, functional connectivity and effective connectivity. For functional activation, we extracted the beta values for each hypoactive ROI as determined in univariate analyses and correlated these values with reading performance. For functional connectivity, we extracted PPI values between connections of interest with reading performance. To find our connections of interest, we first looked at (1) all connections where we found significant differences in coupling between adults with and without dyslexia, and (2) checked whether they included reading-relevant brain regions that we had detected in previous analyses (fMRI-based: left SMG, left vOTC, right vOTC, right cerebellum and bilateral lingual gyri) and classical reading areas from the literature (left posterior STG, left IFG). We ended up with nine connections of interest and correlated the PPI values for these connections with reading performance. To investigate potential effective connectivity-behaviour links, we correlated subject-specific values for intrinsic connectivity between brain regions in both DCMs with reading performance. Concerning reading performance, we used pseudoword and word reading performance from our behavioural assessment and accuracy, reading times and speech onsets for words and pseudoword during fMRI. For all reported brain-behaviour correlations, we computed Bayesian correlations with Kendall's rank correlation coefficient (Kendall's $\tau$) and the Bayesian factor ($BF_{10}$) that quantifies the evidence for H1 as compared to H0 (e.g., a $BF_{10} = 4$ would mean the data is four times more likely under H1 than under H0). Interpretations of $BF_{10}$ are as follows: values of 3–10 provide moderate, 10–30 strong, 30–100 very strong and >100 extremely strong evidence for H1 (vice versa for negative values and H0). We only presented robust correlations with a $BF_{10} > 7$[71] in the manuscript (non-significant findings are additionally presented in Supplementary Tables 14 and 15).

Since data from the present study originates from two neurostimulation studies, the sample size was determined based on comparable previous neurostimulation studies and sensitivity analyses were performed to confirm the chosen sample size was sensitive enough to detect the expected effect size. Earlier neurostimulation studies that aimed to modulate reading performance in dyslexia had reported strong stimulation effects with Cohen's $d$ ranging from −0.37 to 1.96 based on repeated measures ANOVAs with ten subjects[72,73]. Similarly, work from our own group revealed strong stimulation effects after left TPC stimulation in a group of 26 healthy adult subjects (Cohen's $d = 0.63$)[74]. G-power was used to perform a sensitivity calculation, which reveals the smallest effect that could have been detected with high probability given our sample size. The sensitivity analysis showed that assuming α = 0.05, we had 80% power to detect effect sizes larger than 0.55 (Cohen's $dz$) for two-tailed $t$-tests and larger than 0.46 (Cohen's $d$) for repeated measures ANOVAs[75].

**Functional neuroimaging**. MRI data were collected on a 3 T Siemens Magnetom Skyra scanner (Siemens, Erlangen, Germany) with a 32-channel head coil. Blood oxygenation level-dependent (BOLD) images were acquired with a gradient-echo EPI sequence (repetition time [TR]: 2 s, echo time [TE]: 22 ms; flip angle: 80°; field of view [FoV]: 204 mm; voxel size: 2.5 × 2.5 × 2.5 mm; bandwidth: 1794 Hz/Px; phase encoding direction: A/P; acceleration factor: 3). B0 field maps were acquired for susceptibility distortion correction using a spin-echo EPI sequence (TR: 8 s; TE: 50 ms; flip angle: 90°; bandwidth: 1794 Hz/Px). Structural T1 images were taken from the participant database for functional analyse.

We chose a mini-block design with a reading task comprising four conditions – simple words, complex words, simple pseudowords and complex pseudowords (see Stimuli). During fMRI, stimuli were presented for 2.5 s and grouped in mini blocks of 5 stimuli (Fig. 1a). Subjects' task was to read the items they saw aloud. We jittered the inter-stimulus-interval (2.5–4 s) and the rest time, i.e., the interval between mini blocks (7–12 s). Each mini-block lasted for 27.5 s resulting in a total scanning time of 25 min. Subjects were instructed to overtly read stimuli as fast and accurately as possible while avoiding head movements. To record subjects' responses during MRI, an Optoacoustics FOMRI-III dual-channel microphone system was used. Subjects' in-scanner responses were recorded and manually preprocessed using *Audacity*[76]. Speech on- and offsets were determined using *Praat*[77] by four independent raters and 50% of all responses were rated by two of these four raters. We calculated an interrater reliability of >0.85, suggesting high reliability.

We used an event-related mini-block design in which subjects overtly read 100 words (50 simple, 50 complex) and 100 pseudowords (50 simple, 50 complex). The simple, 2-syllable (4–6 letters) words were those provided by Schuster et al.[78]. Complex words comprised the 100 most frequent 4-syllabic German words (10–14 letters) from the *dlex* database (http://www.dlexdb.de/). We first excluded compound words and plurals but included some 3-syllabic words and plurals due to a lack of 4-syllabic words that were neither compounds nor plurals. Pseudowords were generated using the *Wuggy* software (http://crr.ugent.be/programs-data/wuggy) based on the simple and complex word lists. Thus, all pseudowords followed the phonotactic rules of German and no stimulus was repeated. Stimuli were randomized in blocks and blocks were randomized within the run.

**fMRI analysis**. Preprocessing was performed using fMRIprep (20.2.1)[79]. Anatomical T1-weighted images were corrected for intensity non-uniformity (using N4BiasFieldCorrection from ANTs 2.3.3[80], skull-stripped (using antsBrainExtraction from ANTs 2.3.3), segmented into gray matter, white matter and cerebrospinal fluid (using fast in FSL 5.0.9)[81], and normalized to MNI space (MNI152NLin2009cAsym; using antsRegistration in ANTs 2.3.3). Brain surfaces were reconstructed using reconall (FreeSurfer 6.0.1)[82].

Functional BOLD images were co-registered to the anatomical image (using bbregister in FreeSurfer 6.0.1), distortion corrected based on B0-fieldmaps (using 3dQwarp in AFNI 20160207)[83], slice-timing corrected (using 3dTshift from AFNI 20160207), motion corrected (using mcflirt from FSL 5.0.9), normalized to MNI space (via the anatomical-to-MNI transformation), and smoothed with a 5 mm³ FWHM Gaussian kernel (using SPM12 http://www.fil.ion.ucl.ac.uk/spm; Wellcome Trust Centre for Neuroimaging). Moreover, physiological noise regressors were extracted using the anatomical version of CompCor (aCompCor)[84].

We performed a whole-brain random-effects group analysis based on the general linear model (GLM), using the two-level approach in SPM12. First, individual participant data were modelled separately. The participant-level GLM included regressors for the four experimental conditions (simple words, complex words, simple pseudowords, complex pseudowords), modelling trials as box car functions (2.5 s duration) convolved with the canonical hemodynamic response function (HRF). Only correct trials with given answers (incorrect trials were trials without answers or with completely wrong words/pseudowords) were analyzed, error trials were modelled in a separate regressor-of-no-interest. To control for movement artifacts, we included the motion parameters from realignment into the subject-level GLM. To further improve motion confound regression, we also added the motion parameters' temporal derivatives, quadratic terms, and temporal derivatives of the quadratic terms. Therefore, nuisance regressors included 24 motion regressors (the 6 base motion parameters + 6 temporal derivatives of the motion parameters + 12 quadratic terms of the motion parameters and their temporal derivatives)[85,86]. Moreover, we performed motion scrubbing to remove individual time points with strong volume-to-volume movement from the analysis[87]. To this end, we computed framewise displacement (FD) as a measure of excessive volume-to-volume movement and added individual regressors for volumes that exceeded a threshold of FD > 0.9, as proposed for task-based fMRI data[88] (Supplementary Table 16). Finally, we included the top 10 aCompCor regressors explaining the most variance in physiological noise[84]. The data were subjected to an AR(1) auto-correlation model to account for temporal auto-correlations, and high-pass filtered (cutoff 128 s) to remove low-frequency noise.

Contrast images for each participant were computed at the first level. At the second level, these contrast images were submitted to one-sample or paired $t$-tests (to test for interactions). For all second-level analyses, a gray matter mask was applied, restricting statistical tests to voxels with a gray matter probability >0.1 (MNI152NLin2009cAsym gray matter template in fMRIprep). All activation maps were thresholded at a voxel-wise $p < 0.001$ and a cluster-wise $p < 0.05$ FWE-corrected.

Studies using simulations suggest that multivariate pattern analysis (MVPA) is sensitive to the magnitude of variability found on the voxel level (i.e., spatial variation), which is a major shortcoming of univariate analyses. Moreover, MVPA can reduce the noise that is inherent in single-voxel observations through the integration of information coming from several noisy sources[89]. Therefore, we complemented our standard univariate analyses with MVPA using The Decoding Toolbox[90] implemented in Matlab (version 2021a). First, we ran a searchlight MVPA, moving a spherical region-of-interest of 5 mm radius through the entire brain[91]. At each searchlight location, a machine-learning classifier (an L2-norm support vector machine; C = 1) aimed to decode between groups. We performed leave-two-participants-out cross-validation (CV), training on the activation patterns from n-1 participants per group and testing on the left-out two participants (i.e., 1 from each group; yielding 26 CV-folds). This procedure was chosen to test whether the classifier indeed learned to correctly classify people with dyslexia and controls without a bias towards either group. Note that we randomly removed 1 participant from the control group to obtain a balanced dataset of 26 participants per group. Activity patterns comprised beta estimates for each mini block of every participant. For statistical inference, we performed a permutation test across the accuracy-minus-chance maps of the different CV-folds (using SnPM13[92]), thresholded at a voxel-wise $p < 0.001$ and a cluster-wise $p < 0.05$ FWE-corrected.

**Functional and effective connectivity analyses.** PPI allows to investigate task-specific changes in the relationship between activity in different brain regions using fMRI data[93] and thus provides us with the opportunity to compare functional connectivity within brain areas in our two groups: adults with dyslexia and controls. PPI reveals task-dependent changes in functional coupling between an ROI and the whole brain, while controlling for task-independent connectivity (correlation) and task-related activation[93–95]. Concerning ROI selection, we chose our own approach for the outlined reasons below. Most studies applying PPI select a single peak voxel and then draw a (usually random) sphere (mostly between 4 and 8 mm) around this peak coordinate to define their ROIs (a few selected examples[96–98]). However, activation patterns often differ substantially between individuals[99], especially in the case of individuals with atypical activation patterns (e.g. people with dyslexia). Therefore, we believed that it was more meaningful to select the most active voxels in each individual subject, which has also been shown to yield higher sensitivity and functional resolution (i.e., the ability to separate adjacent but functionally distinct regions) than the classical approach of defining ROIs based on the same location in standard space[100]. Initially, we aimed to use an ROI size selection of 10% of the most active voxels as suggested by Fedorenko and colleagues[101,102] and also applied in studies of our own group[103]. However, this ROI size definition led to a very small number of voxels in each cluster since we chose the ROIs based on the hypoactive clusters from the whole-brain univariate analyses. Using only 10% most active voxels yielded 64 voxels for left vOTC, 39 voxels for left SMG, 12 voxels for right vOTC and 9 voxels for the right cerebellum. Clusters of 9–12 voxels are much smaller than a typical seed ROI for PPI, which is why we increased the percentage of included active voxels to achieve clusters of at least 20 voxels in each ROI, resulting in the top 25% of active voxels within regions. Although this is still smaller than in many studies, as summarized in a recent meta-analysis[104], we believe that it is superior to the 10% selection in the present case. To show that this does not substantially change our results, we also provide the PPI results of the respective ROI selection of 10% most active voxels (Supplementary Fig. 14).

In the current study, we employed generalized PPIs[105] to explore task-dependent functional coupling with those brain areas that showed significant hypoactivation in the group of adults with dyslexia in the univariate fMRI analyses. Our seed ROIs were the top 25% most active voxels[101] in the four regions exhibiting largest hypoactivation in adults with dyslexia in our univariate analyses ([CG vs. DYS: all trials > rest]): (1) left SMG (2) left vOTC, (3), right vOTC and (4) right cerebellum. Please note that the cluster in the bilateral lingual gyri was spread across the two hemispheres and the top 25% of most active voxels yielded several smaller clusters spread across the ROI. Therefore, we did not perform PPI with the bilateral lingual gyri.

We performed a whole-brain random-effects group analysis based on the standard two-level generalized linear model (GLM) approach. At the first level, individual subject data were modelled separately using the gPPI toolbox (version 13.1; https://www.nitrc.org/projects/gppi). The participant-level GLM included the following parameters: (1) psychological regressors for experimental conditions (i.e., box car functions convolved with the canonical HRF). Only correct trials were included, while error trials were modelled in a separate regressor of no interest; (2) physiological regressors comprising the first principle component (i.e, eigenvariate) of the time series across voxels in the ROIs; (3) PPI regressors for each experimental condition created by multiplying the deconvolved BOLD signal of the seed ROI with the condition onsets and convolving with the canonical HRF[94];

(4) nuisance regressors as in our activation-based GLM (24 motion regressors, individual regressors for strong volume-to-volume motion, 10 aCompCor regressors).

Contrast images were computed for each subject and subjected to *t*-tests at the second (group) level. We compared connectivity for pseudowords vs. rest, words vs. rest, and words vs. pseudowords (both directions). For all group-level analyses, a grey matter mask was applied, restricting statistical tests to voxels with a grey matter probability >0.1. All activation maps were thresholded at a voxel-wise $p < 0.001$ and a cluster-wise $p < 0.05$ FWE-corrected.

Although PPI can reveal task-dependent changes in functional coupling between a seed region and the rest of the brain, it cannot assess the direction of information flow between brain regions. In other words, it cannot reveal from which of the two connected areas the difference in coupling originates. Consequently, we additionally performed DCM (Friston et al.[106]) to assess directed causal influences (or effective connectivity) between reading-related brain regions[103]. DCM estimates a model of effective connectivity between brain regions to predict a neuroimaging time series[106]. A DCM consists of three types of parameters: (1) intrinsic (condition-independent) directed connections between brain regions (i.e., connections between brain regions that are not influenced by the specific task at hand), (2) modulatory inputs that change connection strengths during a certain experimental manipulation (e.g., words and pseudowords influence intrinsic connections differently, leading to more or less effective coupling from one region to another), and (3) driving inputs that engage network nodes (in our case all written stimuli, i.e., all trials). The goal of DCM is to optimize a tradeoff between model fit (of the predicted to observed time series) and complexity (deviation of model parameters from their prior expectations), measured by the model evidence[107].

We performed a two-level analysis using DCM and a Parametric Empirical Bayes (PEB) framework[108]. At the first level, a full model was specified and estimated for each participant. This full model includes all selected regions and all interactions between these regions. For the first DCM, our full model included the classical reading network from the literature: left TPC, left vOTC, and left IFG. The regions were defined functionally in each individual participant as the top 10% most active voxels for [all trials > rest] within 20 mm spheres around the MNI peak coordinates from the meta-analysis by Martin et al.[109]: left TPC = −49 −44 21; left IFG = −52 20 18; left vOTC = −42 −68 −22. The second DCM was based on a full model that included our extended reading network, i.e., the four most underactivated clusters (see PPI): left SMG, left vOTC, right vOTC and right cerebellum. These ROIs were defined functionally in each participant as the top 25% of most activated voxels for the contrast [all trials > rest]. The full models assumed that all ROIs were fully connected via reciprocal connections. We set the onset of all stimuli as driving input to all ROIs, and words and pseudowords as modulatory inputs on the connections between ROIs. The first eigenvariate of the BOLD time series of each region was extracted and adjusted for effects of interest (all experimental conditions) using our participant-level GLM. DCM inputs were mean-centered, so that the intrinsic connections reflected the mean connectivity across experimental conditions[107].

At the second level, DCM parameters of individual participants were entered into a GLM—the PEB model—that decomposed interindividual variability in connection strengths into group effects and random effects[110]. We then compared the full model against 256 reduced models that had parameters switched off that did not contribute to the model evidence (i.e., prior mean and variance set to 0)[108]. For group-level inferences, one can choose an optimal model that is identical across a population or compute a Bayesian Model Average (BMA), the average of parameter values across models weighted by each model's posterior probability of competing models (Pp)[111,112]. We chose the BMA approach exclusively assessing the parameters of the best model as the BMA accommodates uncertainty about the true underlying model[108]. The BMA was thresholded to only retain parameters with a Pp > 95%[110], meaning that we only report parameters with posterior probabilities bigger than 0.95 on the group level as significant. For each modulatory input, we calculated the resulting connectivity value (in Hz) using formula 3[110]. Finally, we directly compared different parameters on the same connection using Bayesian contrasts.

**Reporting summary.** Further information on research design is available in the Nature Portfolio Reporting Summary linked to this article.

## Data availability

Data underlying all analyses and figures is provided via the following OSF registry: https://osf.io/cy8tk/[113]. No password or registration is needed to access the data. The filenames within the data repository structure provide information as to which figure and analysis the data was used for. For statistical analyses, not only the data tables but JASP and R files were uploaded as well.

## Code availability

All code is available via OSF registry: https://osf.io/cy8tk/[113]. This includes the specific neuroimaging code and all resulting .nii files that were plotted to obtain the figures presented in the manuscript and the supplements. Additionally, code for the linear mixed models is provided together with the respective datasheet within its own folder.

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

## Acknowledgements

We thank all subjects for their participation. The present work was supported by the Alexander von Humboldt Foundation, the Lise Meitner Excellence Program of the Max Planck Society, the German Research Foundation and the European Research Council.

## Author contributions

S.T.: conceptualization, methodology, investigation, writing—original draft, visualization, project administration, funding acquisition. P.K.: software, data curation, formal analysis, writing—review & editing. Z.J.: formal analysis. G.H.: resources, supervision, funding acquisition, writing—review & editing.

## Funding

## Competing interests

The authors declare no competing interests.
