## [Peer Review File · Communications Biology]

Reviewers' comments:

Reviewer #1 (Remarks to the Author):

This is a very interesting report of reading related activation and FC disturbances in adult persons with reading difficulties. The authors creatively integrate conventional event-related fMRI activation contrast with MVPA, connectivity-performance correlations and effective connectivity analyses. They identify activations and connectivity features, some of which are expected based on the rather extensive research in this field, and some less expected, including a central role of the cerebellum and spl in the network of areas supporting poorly developed reading skills. Given that the import of these somewhat unexpected findings, I strongly suggest some more conventional/straightforward analyses to further support the overall generalizability of their findings (especially the more novel ones). Specifically: (1) Perform correlational analyses (preferably bivariate) between b values (extracted from 1st level contrasts of e.g. complex W>rest the regions identified by the corresponding second level contrast) with appropriate performance indices (in-scanner word accuracy/latency and out of the scanner word reading accuracy) separately in each group (with emphasis on RD); (2) Along the same lines perform connectivity-performance correlations in a more basic (yet perhaps more interpretable approach). You could: (I) determine individual seed regions showing significant e.g. complex word>rest activations for instance in L SMG and L OTC, (Ii) extract average waveforms from each region during e.g., complex word reading, (iii) compute z-transformed correlations between the two waveforms and (iv) compute correlations between z values and word reading accuracy/latency in-scanner and word reading accuracy out of the scanner. (3) I am not sure that the mvpa results add much to the overall picture. As I understand, mvpa clusters are not used in further analyses. Also I wonder if the number of usable events for complex pseudowords among some dyslexics.

Additional issues to be addressed are detailed below. How typical is the sample of RD adults of the respective population? They are obviously partly compensated dyslexics as indicated by average text reading accuracy. What about special education history? Vocabulary scores to help explain poor text comprehension in the presence of average reading accuracy? Were there more persons screened to end up with the current sample? Are scores in table 1 standard? Percentage of persons scoring <1.5 sd's below the population mean?

Scanning parameters: reason for using a very short TE? Longer TE values are known to lead to better snr.

In addition to motion regressions what other steps were taken to ensure that excessive, systematic head movement did not affect the quality of the data? Incidentally the explanation of the motion correction in lines 524-5 is a bit sketchy. Was there a threshold set for max displacement on any of the 3axes?

Please add time scale to fig 1-top panel. What was the length of the rest window (constant during the variable interstimulus interval)?

Group activation maps of reading vs rest for each of the four conditions would be highly informative (not just tables of activations for word and pseudo word vs rest presented in supplementary material especially in view of the extensive differences between simple and complex stimuli shown in fig 2.

The extensive common activations for word>pseudowords and for simple>complex words mostly in dmn areas are curious. Have they been consistently reported in previous research in similar contrasts? Any chance they indicate that the glm model was not successful in removing a lot of the background hemodynamic activity? If that were the case could it account for the connectivity findings involving spl?

Methods. Lines 558-565 are somewhat confusing as the DCM approach has not been mentioned earlier. Also given the complexity of the analysis pipeline it would help to describe the ppi and dcm analyses step by step, explaining what the dependent variable is at each step (and which indices computed at earlier steps are used in later ones, if applicable).

Reviewer #2 (Remarks to the Author):

The main objective of this study is to investigate the potential relationship between dyslexia and disruptions in brain networks. The authors applied various techniques, such as functional and effective connectivity, to examine this hypothesis. Their findings indicate that impaired connectivity between reading regions and other regions may contribute to dyslexia, emphasizing the significance of the right cerebellum in dyslexia. Additionally, the authors demonstrated a correlation between connectivity patterns and behavior, revealing that stronger connectivity was associated with poorer reading abilities in individuals with dyslexia.

Major comments

1) My main comment is concerning the lack of detail and description in the Figures, Methods, and Results sections. In its current form, it is hard to understand what specific analysis the authors conducted and for what reason. Several different analyses were conducted but the manuscript does not convey the reasoning behind these analysis choices.

To give just a few examples:

L 140-141: It is not well motivated why the authors choose to use MVPA. Regarding this experiment what can MVPA reveal that cannot be achieved by univariate analysis? Please explain.

L149: How would the authors explain that in Figure 3 regions that are part of the "classical reading network" (IFG and TPC as indicated in L184) are not significant in the univariate and MVPA analysis results?

L 461: "Please note that only data from the ineffective sham (placebo) stimulation session entered the here reported analyses." The authors do not describe why this data was included and why the rest of the data was excluded from the analysis.

L571, L602, L607: I cannot identify any systematic search that is conducted when the number of voxels that are included in the PPI and DCM analysis are selected. 25%, 10% or 10% most active voxels are selected in different analyses. Is there a particular reason that these numbers were selected? How reliable are the results when a varying number of voxels were chosen?

L573: Which deconvolution method did the authors use?

L617: How many reduced models and comparisons were made? Please also specify which "certain parameters" were switched off.

Regarding Figure captions, I suggest the authors review all figure captions. In its current form, the figure captions lack the necessary details to understand the figure. For example, a good figure caption would include a brief description of the methods and statistical information necessary to understand the figure without having to refer to the main text.

2) The authors mainly show subject-averaged results, but individual subject variability in language processing needs to be considered. The authors do acknowledge "large individual variability in the dyslexia group" (L555-556) but do not show individual subject results. It would be important to show the evidence that results replicate in each individual subject (Fedorenko 2012).

Reference:

Fedorenko E, Nieto-Castañón A, Kanwisher N. Syntactic processing in the human brain: what we know, what we don't know, and a suggestion for how to proceed. *Brain Lang.* 2012 Feb;120(2):187-207.

3) L83: I find the statement "neural marker" (also in the title) rather strong given the complexity of the processes and their yet unresolved nature.

4) L 484-486: "Subjects were instructed to overtly read stimuli as fast and accurately as possible while avoiding head movements."

Head motion during overt reading can strongly affect the SNR. How much motion was visible in each scan session? How was head movement controlled / "avoided"? Did the authors assess how SNR is affected with and without overt reading?

Also, was a noise-canceling microphone used to process the in-scanner overt responses?

5) L570: I suggest citing the original papers when describing PPI:

Friston KJ, Buechel C, Fink GR, Morris J, Rolls E, Dolan RJ. Psychophysiological and modulatory interactions in neuroimaging. *Neuroimage.* 1997; 6:218-229.

Gitelman DR, Penny WD, Ashburner J, Friston KJ. Modeling regional and psychophysiological interactions in fMRI: the importance of hemodynamic deconvolution. *Neuroimage.* 2003; 19:200207.

Minor comments

1) L10: "Our results reveal disrupted functional coupling between hypoactive reading regions and large clusters across the brain, with the right cerebellum showing the largest disruption": Please specify "large clusters across the brain".

2) L22-23: "Stronger connectivity within networks is linked to worse reading performance" Specify to which subject group this finding refers to.

3) L 84, Results: To make the results section more comprehensible, I suggest the authors include one paragraph at the beginning of the section that describes the specific experimental design referring to Figure 1, the tasks, and study goals. This will help to better/faster understand L96-102.

4) L87-90: Table 1: A short description of the tasks included as supplemental material would be helpful.

5) L127, Figure2: The colormap used to visualize "Simple vs. Complex" conditions is very hard to distinguish. I suggest changing this to a different colormap with stronger color contrast.

6) L179: Figure 4A and 4C: Why are the bar plots only highlighted for the connectivity between L vOTC and L/R cuneus?

7) L326: Typo in "showws reduced functional [...]"

8) L377: Typo "we" in "Interestingly, we the direct group comparison also showed increased coupling from [...]."

9) L458: Describe which specific behavioral tests were used.

10) L501-506: Please cite the corresponding manuscripts for ANT, Freesurfer, AFNI, and SPM.

11) L518: It would be useful to list the four experimental conditions here. In Methods and Figure 1, these conditions are mentioned but not explicitly. I would also suggest making the four experimental conditions in L491-492 in Methods explicit.

12) L540: Was there a specific reason why a "leave-two-participants-out" procedure was chosen over the more common "leave-one-participant-out" procedure?

Dear Reviewers,

Thank you for your positive and constructive feedback. Please find our detailed responses to your feedback (in *italics*) under the corresponding comment / paragraph below. Excerpts from the manuscript are shown with quotation marks and indented.

Reviewer 1:

This is a very interesting report of reading related activation and FC disturbances in adult persons with reading difficulties. The authors creatively integrate conventional event-related fMRI activation contrast with MVPA, connectivity-performance correlations and effective connectivity analyses. They identify activations and connectivity features, some of which are expected based on the rather extensive research in this field, and some less expected, including a central role of the cerebellum and spl in the network of areas supporting poorly developed reading skills.

Thank you for your positive feedback!

Given that the import of these somewhat unexpected findings, I strongly suggest some more convectional/straightforward analyses to further support the overall generalizability of their findings (especially the more novel ones). Specifically: (1) Perform correlational analyses (preferably bivariate) between b values (extracted from 1st level contrasts of e.g. complex word>rest the regions identified by the corresponding second level contrast) with appropriate performance indices (in-scanner word accuracy/latency and out of the scanner word reading accuracy) separately in each group (with emphasis on RD); (2) Along the same lines perform connectivity-performance correlations in a more basic (yet perhaps more interpretable approach). You could: (I) determine individual seed regions showing significant e.g. complex word>rest activations for instance in L SMG and L OTC, (Ii) extract average waveforms from each region during e.g., complex word reading, (iii) compute z-transformed correlations between the two waveforms and (iv) compute correlations between z values and word reading accuracy/latency in-scanner and word reading accuracy out of the scanner.

Thank you for your constructive feedback and ideas. Following your suggestions, we performed correlational analyses between beta values in our five regions-of-interest (ROIs) that showed underactivation in people with dyslexia (as compared to controls) and in- and out-of-scanner performance indices. We found strong activation-behaviour correlations when considering both groups. However, please note that we found no significant correlations when considering each group in isolation (i.e., within the dyslexia or the control group, brain activation is not correlated with behaviour). This suggests that the activation-behaviour correlations might have been mainly driven by group differences or, in other words, by large differences in performance when considering both groups. If we ignore the group / diagnostic status and think about reading as a continuum, brain activation seems to reflect differences in reading ability, with the left SMG showing highest correlation with pseudoword reading performances. We have added all correlations for in- and out-of-scanner performance with functional activation in an extensive table in the Supplements (**Table S14**) and we additionally report significant correlations in the manuscript (**Table 2, see below**) (see **Section 2.3.** from page 15 on). Please note that plots for all significant correlations can also be found in the Supplements (**Figures S3-S7**).

“In terms of functional activation, we found strong correlations between activation in the hypoactive brain areas (left SMG, left vOTC, right vOTC, right cerebellum and bilateral lingual gyri) and both reading performance (in and outside the scanner) across groups. Significant correlations are reported in **Table 2**, all results, including non-significant results, are displayed in the **Supplemental Table S14**. These correlations suggest that the left SMG and the right cerebellum show strong pseudoword-specific recruitment, whereas left and right vOTC and bilateral lingual gyri show stronger links to word reading (see **Table 2** for concrete values). Please note that these strong correlations were not significant within groups. Correlational plots are also provided in **Supplemental Figures S3-S7**.

Table 2. Bayesian correlations for functional activation and reading performance. We found correlations for activation within the five hypoactive ROIs from the univariate analyses (CG > DYS [all trials > rest]), namely left vOTC, left SMG, right vOTC, right cerebellum and bilateral lingual gyri, with in- and out-of-scanner performance. Out-of-scanner performance comprised word and pseudoword reading as assessed by the SLRT-II, whereas in-scanner performance included pseudoword / word speech onsets, reading times and accuracy. Whereas the right cerebellum was only linked to pseudoword processing, all other areas showed a link to both types of stimuli, with stronger effects to one of the two stimulus types, however. We report Kendall’s τ for robust correlations with a $BF_{10} > 7$ ³².

ROI	Performance	τ	BF₁₀
L vOTC	pseudoword reading (SLRT)	0.323	52.775
	pseudoword reading times (MRI)	-0.273	9.748
	pseudoword accuracy (MRI)	0.304	24.109
	word reading (SLRT)	0.395	911.539
L SMG	pseudoword reading (SLRT)	0.497	48686.612
	pseudoword speech onsets (MRI)	-0.309	30.107
	pseudoword reading times (MRI)	-0.422	2509.199
	pseudoword accuracy (MRI)	0.334	71.321
	word reading (SLRT)	0.404	1334.189
	word speech onsets (MRI)	-0.27	8.93
R vOTC	pseudoword reading (SLRT)	0.309	33.531
	pseudoword accuracy (MRI)	0.272	9.341
	word reading (SLRT)	0.344	116.502
R cerebellum	pseudoword accuracy (MRI)	0.304	25.109
B lingual	pseudoword reading (SLRT)	0.302	26.275
	word reading (SLRT)	0.347	364.659
	word speech onsets (MRI)	-0.262	7.203

Considering our connectivity analyses, we believe that the psychophysiological interaction (PPI) analyses we have performed are superior to simple correlations between timeseries (or “waveforms”) as suggested. Specifically, in task-based fMRI studies (like the present one), psychophysiological interaction (PPI) analyses are preferred over simple timeseries correlations as PPI controls for effects of task-independent connectivity and common task-related activation (O’Reilly et al., 2012). In other words, PPI enables us to identify brain regions that exhibit task-dependent functional connectivity with a certain seed region-of-interest (ROI), above and beyond their task-independent connectivity (correlation) and task-related activation. Nonetheless, we agree that it does make sense to correlate our PPI findings

with our behavioural results. Therefore, we extracted PPI values for clusters between our areas of interest (left SMG, left pSTG, left vOTC, right vOTC, right cerebellum) for [words > rest] and [pseudowords > rest] contrasts that showed significant differences in coupling between the two groups. In total, we extracted values for each participant for nine connections of interest and correlated them with the respective in-scanner and out-of-scanner performance. Significant results are shown in **Table 3**, whereas the extensive table including all correlations can be found in the Supplements (**Table S15**).

“In a next step, we extracted PPI functional connectivity values for each single participant and correlated them with in- and out-of-scanner reading performance across group. Since we found less functional coupling in numerous areas in the dyslexia group in our PPI analysis (see **Figure 5**), we decided to limit the correlations to functional connections between areas that we designated as classical and extend reading network nodes in our DCM analysis (see **Figure 7**). These comprised nine connections of interest starting always with the respective seed region: (1) left SMG - left vOTC, (2) left vOTC - left SMG, (3) left SMG - right cerebellum, (4) right cerebellum - left vOTC, (5) left vOTC - right vOTC, (6) right cerebellum - left temporal cortex (the cluster as found in the PPI comprised large portions of left pSTG and pMTG), (7) right cerebellum - left IFG, (8) right cerebellum - left vOTC and (9) right cerebellum - right vOTC. Bayesian correlations with a $BF_{10} > 7$ are reported in **Table 3**, the whole range of results, including non-significant findings, are provided in **Supplemental Table S15**.

Table 3. Bayesian correlations for connections of interest derived from PPI analyses and reading performance. Out-of-scanner performance comprised word and pseudoword reading as assessed by the SLRT-II, whereas in-scanner performance (MRI) included pseudoword / word speech onsets, reading times and accuracy. Whereas correlations with word reading comprised only out-of-scanner performance and specifically the left SMG to left vOTC coupling, pseudoword performance was tied to connections originating from the right cerebellum. We report Kendall’s τ for robust correlations with a $BF_{10} > 7$ ³².

Connections of interest (PPI)	Performance	τ	BF_{10}
L SMG - L VOTC	Word reading (SLRT)	0.347	130.044
L vOTC - L SMG	Word reading (SLRT)	0.355	171.881
L SMG - R Cerebellum	Word reading (SLRT)	0.304	27.530
R Cerebellum - L vOTC	Word reading (SLRT)	0.298	22.736
R Cerebellum - L Temporal	Pseudoword accuracy (MRI)	0.270	8.936
R Cerebellum - L IFG	Pseudoword reading times (MRI)	-0.305	25.950
R Cerebellum - R vOTC	Pseudoword reading (SLRT)	0.266	8.468

Connectivity between left SMG and left vOTC was strongly tied to word reading performance ($\tau = 0.347$, $BF_{10} = 130$; $\tau = 0.355$, $BF_{10} = 171.8$) during our behavioural assessment, both when the left SMG and the left vOTC were taken as seeds in the

PPI. Moreover, word reading performance outside the scanner was tightly linked to connectivity between left SMG and the right cerebellum ($\tau = 0.304$, $BF_{10} = 27.5$), as well as the right cerebellum and left vOTC ($\tau = 0.298$, $BF_{10} = 22.7$). Connectivity from the right cerebellum to three other regions (left temporal: $\tau = 0.270$, $BF_{10} = 8.9$; left IFG: $\tau = -0.305$, $BF_{10} = 26$; right vOTC: $\tau = 0.266$, $BF_{10} = 8.5$), in contrast, correlated strongly with pseudoword reading performance (in and out of the scanner). Higher connectivity values always were always associated with better performance. When we performed the same correlations within each group only, we did not find any significant correlations. Looking at reading as a continuum, however, our results suggest that better word reading performance is linked to overall stronger functional interaction between left SMG and left vOTC. Better pseudoword reading performance, in contrast, was linked to stronger connectivity between the right cerebellum and bilateral reading-related areas (left IFG, left temporal cortex, right vOTC) (see **Supplemental Figures S6 and S7**). Please note that these findings were only significant when looking at reading ability as a continuum and including both groups, but not within groups and should thus be interpreted with caution.”

(3) I am not sure that the mvpa results add much to the overall picture. As I understand, mvpa clusters are not used in further analyses. Also I wonder if the number of usable events for complex pseudowords among some dyslexics.

MVPA is a reliable method to detect differences in activation patterns that often go unnoticed in standard univariate analyses. Therefore, we believe that using MVPA to complement our univariate analyses is important and can help detect regions that were not found with standard analyses. We agree that we did not adequately discuss our MVPA findings in the results and discussion section, but rather used them to confirm our univariate findings. We have added the following paragraph in the results and discussion section to address this (page 19, from line 418):

“We confirm that dyslexia is characterized by marked hypoactivation in classical reading areas. Our univariate findings support a difference in left-hemispheric recruitment for words and pseudowords. The left-hemispheric hypoactivation in the left TPC and OTC corroborates earlier findings of meta-analyses on functional activation in dyslexia^{14,15,40}. Interestingly, however, our results of the right cerebellum stand in contrast to recent work linking it to compensation in shallow orthographies⁴¹ and as functional compensatory marker for dyslexia in children⁴². In the present study, we did not detect compensatory activation as suggested by other studies^{15,31,42}. MVPA further identified additional fine-grained differences in functional activation patterns in domain-general regions across the two hemispheres. These regions included the left insula, the bilateral anterior cingulate and the right precuneus. Whereas a dysfunctional insula was suggested as a marker for dyslexia already in early studies⁴³ and is supported by a recent meta-analysis⁴⁴, anterior cingulate activation is usually linked to increased attention and effort⁴⁵ and negatively linked to reading performance⁴⁶. In earlier studies, the right precuneus was less activated in adults with dyslexia when compared to controls during working-memory-related tasks^{47,48}. Activation differences in the precuneus and cingulate gyri could thus be interpreted as deficits in fully engaging domain-general processing regions that support reading processes vital for reading skills in dyslexia⁴⁹. Last, we also found differences in activity patterns of the left cerebellum. Activation in the

cerebellum has been implicated in higher cognitive functions and executive control, and has recently been linked to semantic processing^{23,50}.”

Concerning your second comment on the number of usable events for complex pseudowords among some adults with dyslexia, we would like to add that incorrect responses (which we kept as part of the analyses) were almost exclusively small reading mistakes. In other words, when subjects didn't say anything or read something totally wrong, we did exclude these instances. When they only exchanged two letters (e.g., Edxersprine instead of Exdersprine) or left out a single sound / letter, we did not drop these trials. Since we were hoping to capture the reading process, we believe that leaving out all trials with such small mistakes would not accurately reflect and capture the reading process in dyslexia. Even if there were such tiny mistakes, they still read the letters on the screen and came up with pseudowords using sound-symbol conversion processes. Therefore, the number of complex pseudowords for the subjects with dyslexia was sufficient to allow for a comparison across groups and conditions.

Additional issues to be addressed are detailed below. How typical is the sample of RD adults of the respective population? They are obviously partly compensated dyslexics as indicated by average text reading accuracy. What about special education history? Vocabulary scores to help explain poor text comprehension in the presence of average reading accuracy? Were there more persons screened to end up with the current sample? Are scores in table 1 standard? Percentage of persons scoring <1.5 sd's below the population mean?

Thank you for pointing out this important issue. We think that the sample of adults with dyslexia is as typical as possible – especially given that there is no typical or homogenous sample of adults with dyslexia. We only included adults with a prior diagnosis and partial compensation, most had been to dyslexia classes from age 7-8, which is the standard case in Germany for severe cases of dyslexia. These classes are taken at the age of 7 and include additional reading training while doing one typical school year spread across two years. In essence, they are all partly compensated dyslexics, who, however, still struggle with almost all reading tasks that we administered. The text reading accuracy score did not significantly differ between the two groups because adults with dyslexia just took much longer and checked each response more often to make sure they had the correct answer. This explains why their accuracy was very high, but their speed was very low (a trade-off). Therefore, we would not over-interpret this score.

Concerning special education history, we are currently doing a qualitative sample with most of the adults with dyslexia from that study, trying to better capture the overall picture. This is based on interviews, however, and will take more time to be analysed and interpreted accordingly. For the present study, we can say that there was a lot of individual variability in individual educational paths. None of our subjects went to a special education school but most did attend dyslexia classes. Moreover, almost all of them were students or had already completed their studies (this is the sample we primarily get in our research institute).

Regarding your question about the drop-out rate for the current study, we only had to exclude two participants from the dyslexia group. One was excluded due to excessive movement during scanning, the other scored above average compared to the control group (despite prior history of reading and spelling problems). We had such a low drop-out rate because we already screened participants excessively before inviting them – making sure they were allowed to do MRI, no prior history or diagnosis of AD(H)D or dyscalculia, no left-handedness, a prior diagnosis of dyslexia or a detailed description of their history of reading and spelling problems. Only if they met these criteria, they were invited to the behavioural

session and then we made sure to only include suitable participants (i.e., who met our criteria of scoring at least in half of the tests 1.5 standard deviations below the mean of the control group) to the fMRI sessions. Still, this does not mean that there was no large individual variability in performance. However, we believe that it is impossible to get a homogenous sample of any clinical or ‘atypical’ population, especially in the case of adults.

The scores in Table 1 are the concrete scores they received during the testing. Please see our extended behavioural assessment description in the methods for details. W-value – and p-value show the (significant) differences between the groups.

Scanning parameters: reason for using a very short TE? Longer TE values are known to lead to better snr.

In addition to motion regressions what other steps were taken to ensure that excessive, systematic head movement did not affect the quality of the data? Incidentally the explanation of the motion correction in lines 524-5 is a bit sketchy. Was there a threshold set for max displacement on any of the 3axes?

The TE was chosen after consultation with our neurophysics department and as a compromise to balance BOLD sensitivity and signal dropout: while longer TE values are indeed associated with higher BOLD sensitivity in superior brain regions, they are also associated with more severe signal dropout in inferior areas (Poser et al. 2006).

Regarding motion correction, we included the motion regressors from realignment to our subject-level general linear model (GLM), as well as their temporal derivatives, quadratic terms, and temporal derivatives of the quadratic terms. This yielded 24 motion regressors in total. Moreover, we calculated framewise displacement (FD) to identify excessive volume-to-volume movement and excluded any volumes exceeding a threshold of $FD > 0.9$, as suggested by Siegel et al. (2014) for task-based fMRI data. This procedure, also known as “motion scrubbing”, was achieved by adding single-volume regressors for these high-movement volumes to the subject-level GLM. Finally, we quantified physiological noise using the anatomical version of CompCor (aCompCor) and included the top 10 regressors explaining the most variance. This information now appears in the Methods section (p. 28-29):

“We performed a whole-brain random-effects group analysis based on the general linear model (GLM), using the two-level approach in *SPM12*. First, individual participant data were modeled separately. The participant-level GLM included regressors for the four experimental conditions (simple words, complex words, simple pseudowords, complex pseudowords), modelling trials as box car functions (2.5 seconds duration) convolved with the canonical hemodynamic response function (HRF). Only correct trials with given answers (incorrect trials were trials without answer or with completely wrong words / pseudowords) were analyzed, error trials were modeled in a separate regressor-of-no-interest. To control for movement artifacts, we included the motion parameters from realignment into the subject-level GLM. To further improve motion confound regression, we also added the motion parameters’ temporal derivatives, quadratic terms, and temporal derivatives of the quadratic terms. Therefore, nuisance regressors included 24 motion regressors (the 6 base motion parameters + 6 temporal derivatives of the motion parameters + 12 quadratic terms of the motion parameters and their temporal derivatives)^{78,79}. Moreover, we performed “motion scrubbing” to remove individual time points with

strong volume-to-volume movement from the analysis⁸⁰. To this end, we computed framewise displacement (FD) as a measure of excessive volume-to-volume movement and added individual regressors for volumes that exceeded a threshold of $FD > 0.9$, as proposed by Siegel for task-based fMRI data⁸¹. Finally, we included the top 10 aCompCor regressors explaining the most variance in physiological noise⁷⁷. The data were subjected to an AR(1) auto-correlation model to account for temporal auto-correlations, and high-pass filtered (cutoff 128s) to remove low-frequency noise.”

Please add time scale to fig 1-top panel. What was the length of the rest window (constant during the variable interstimulus interval)?

Thank you for pointing this out, this has been added to the respective figure. The rest window was jittered between 7 and 12 seconds, as now more clearly stated also in 5.1 (page 27):

“We chose a mini-block design with a reading task comprising four conditions – simple words, complex words, simple pseudowords and complex pseudowords (see Stimuli). During fMRI, stimuli were presented for 2.5 seconds and grouped in mini blocks of 5 stimuli (**Figure 1A**). Subjects’ task was to read the items they saw aloud. We jittered the inter-stimulus-interval (2.5 - 4 seconds) and the rest time, i.e., the interval between mini blocks (7 - 12 seconds). Each mini-block lasted for 27.5 seconds resulting in a total scanning time of 25 minutes.

Please note that we also added this information in the description of Figure 1.

“Figure 1. fMRI study design and behavioural results. (A) During fMRI, subjects read simple and complex words (red) and simple and complex pseudowords (blue). Mini-blocks always contained 5 items and lasted for 27.5 seconds. Inter-stimulus intervals and rest periods were jittered (2.5 - 4 seconds and 7 - 12 seconds respectively). (B) Linear mixed model results show group-differences in task performance, including speech onsets, reading times and reading accuracy. Significant post-hoc tests are indicated (** $p < 0.001$, ** $p < 0.01$, * $p < 0.05$). Adults with dyslexia read all items slower but needed significantly longer to overtly produce pseudowords. Moreover, their accuracy was only significantly lower for the pseudoword reading blocks, with worse performance during complex pseudoword reading.”

Group activation maps of reading vs rest for each of the four conditions would be highly informative (not just tables of activations for word and pseudo word vs rest presented in supplementary material especially in view of the extensive differences between simple and complex stimuli.

We agree and now provide the plots for the four conditions vs. rest in the supplements – see Figures S1 and S2. As you can see, there is practically no visible difference in activation for the four conditions – the major differences when comparing words and pseudowords and complex and simple stimuli stem from deactivation for the respective other condition. In other words, word processing is marked by deactivation in many regions of the default mode network, which then leads to differences in activation when comparing them with pseudowords.

Figure S1. Functional activation results for all four conditions (simple words, complex words, simple pseudowords, complex pseudowords) vs. rest in the control group ($p < 0.001$ voxel-level, $p < 0.05$ cluster-wise FWE corrected).

CONTROL GROUP

simple words > rest

complex words > rest

simple pseudowords > rest

complex pseudowords > rest

Figure S2. Functional activation results for all four conditions (simple words, complex words, simple pseudowords, complex pseudowords) vs. rest in the dyslexia group ($p < 0.001$ voxel-level, $p < 0.05$ cluster-wise FWE corrected).

The extensive common activations for word > pseudowords and for simple > complex words mostly in dmn areas are curious. Have they been consistently reported in previous research in similar contrasts? Any chance they indicate that the glm model was not successful in removing a lot of the background hemodynamic activity? If that were the case could it account for the connectivity findings involving spl?

Most studies in the past have confirmed that directly contrasting words and pseudowords leads to stronger DMN activation for words compared with pseudowords and task-positive network activation for pseudowords (see Binder et al., 2009, for a meta-analysis). Only few studies have found contrary findings (e.g., Mattheiss et al., 2018). However, many studies used a lexical decision task and stimuli not differing in complexity, limiting the comparison between previous studies and our study in terms of stimulus complexity. Nevertheless, we are confident that our results are valid since they nicely converge with the above-mentioned literature. A plausible explanation for the common activation of default mode regions in words > pseudowords and simple > complex stimuli is task difficulty: easier or more automatic tasks (words and simple stimuli) often show more activation (or less deactivation)

of DMN regions (see Humphreys et al., 2015, for a meta-analysis), which would be reflected in more activation in the specific contrasts in our study.

Binder, J. R., Desai, R. H., Graves, W. W., & Conant, L. L. (2009). Where is the semantic system? A critical review and meta-analysis of 120 functional neuroimaging studies. *Cerebral cortex*, *19*(12), 2767-2796.

Humphreys, G. F., Hoffman, P., Visser, M., Binney, R. J., & Lambon Ralph, M. A. (2015). Establishing task-and modality-dependent dissociations between the semantic and default mode networks. *Proceedings of the National Academy of Sciences*, *112*(25), 7857-7862.

Mattheiss, S. R., Levinson, H., & Graves, W. W. (2018). Duality of function: activation for meaningless nonwords and semantic codes in the same brain areas. *Cerebral Cortex*, *28*(7), 2516-2524.

Methods. Lines 558-565 are somewhat confusing as the DCM approach has not been mentioned earlier. Also given the complexity of the analysis pipeline it would help to describe the ppi and dcm analyses step by step, explaining what the dependent variable is at each step (and which indices computed at earlier steps are used in later ones, if applicable).

Thank you for pointing this out. We have extended the methods section to allow for a replication of our analyses in future studies – and of course for a better understanding of the performed analyses. We now also provide more information on the rationale behind these analyses – see **pp. 30-33** in the Methods section.

Reviewer 2:

The main objective of this study is to investigate the potential relationship between dyslexia and disruptions in brain networks. The authors applied various techniques, such as functional and effective connectivity, to examine this hypothesis. Their findings indicate that impaired connectivity between reading regions and other regions may contribute to dyslexia, emphasizing the significance of the right cerebellum in dyslexia. Additionally, the authors demonstrated a correlation between connectivity patterns and behavior, revealing that stronger connectivity was associated with poorer reading abilities in individuals with dyslexia.

Major comments

1) My main comment is concerning the lack of detail and description in the Figures, Methods, and Results sections. In its current form, it is hard to understand what specific analysis the authors conducted and for what reason. Several different analyses were conducted but the manuscript does not convey the reasoning behind these analysis choices.

Thank you for pointing this out – we see your point and have provided more insight into the analyses and the rationale behind choosing the respective analyses (please see Methods section **pp. 25-33** and see below for details on your comments). We hope that you find our methods section now more comprehensive and detailed.

To give just a few examples:

L 140-141: It is not well motivated why the authors choose to use MVPA. Regarding this experiment what can MVPA reveal that cannot be achieved by univariate analysis? Please explain.

As briefly described in the methods section, univariate analyses often fail to detect more subtle differences that are often captured with MVPA. In our case, we found hypoactivation in more classical regions through the univariate analysis, but also confirm earlier reports of hypoactivation only through our MVPA.

We hope this additional information clarifies this (p.29):

“Studies using simulations suggest that multivariate pattern analysis (MVPA) is sensitive to the magnitude of variability found on the voxel level (i.e., spatial variation), and thus more sensitive to variability on the subject-level, which is a major shortcoming of univariate analyses. Moreover, MVPA can reduce the noise that is inherent in single-voxel observations through integration of information coming from several noisy sources (81). Therefore, we complemented our standard univariate analyses with a MVPA using *The Decoding Toolbox* (82) implemented in *Matlab* (version 2021a).”

L149: How would the authors explain that in Figure 3 regions that are part of the “classical reading network” (IFG and TPC as indicated in L184) are not significant in the univariate and MVPA analysis results?

It might be misleading that we did not separately report the (simple/complex) word/pseudoword vs. rest contrasts in the first version of the manuscript because they do show that

the IFG and TPC were crucially engaged during our task in both groups (see Figures S1 and S2).

Figure S1. Functional activation results for all four conditions (simple words, complex words, simple pseudowords, complex pseudowords) vs. rest in the control group ($p < 0.001$ voxel-level, $p < 0.05$ cluster-wise FWE corrected).

Figure S2. Functional activation results for all four conditions (simple words, complex words, simple pseudowords, complex pseudowords) vs. rest in the dyslexia group ($p < 0.001$ voxel-level, $p < 0.05$ cluster-wise FWE corrected).

Please note that the findings presented in Figure 3 are not general activations for the task but significant differences between the two groups. Additionally, differences in functional connectivity do not always necessarily go hand in hand with differences in functional activation. We based our DCM on the classical literature-based reading model, not on the differences between the two groups. Our motivation was the following: left IFG and TPC are vital for reading as they appear highly active in our analyses vs. rest. However, we see your point and that is also why we tried the more extensive reading network, to check if our data supports that it might not only be the classical reading network that explains processing differences.

L 461: "Please note that only data from the ineffective sham (placebo) stimulation session entered the here reported analyses." The authors do not describe why this data was included and why the rest of the data was excluded from the analysis.

Thank you for pointing this out – we just mentioned this to be transparent. The two overall studies (controls and adults with dyslexia) combined non-invasive brain stimulation and fMRI. The first major aim of this 3-year project was to look at a possible neuromodulation through non-invasive brain stimulation, and the other to compare activation and connectivity between groups, also as a basis to interpret effects of stimulation. The present article answers the second question. The other article is currently under revision but does not address the aspects discussed in this manuscript. We have slightly changed this and have also added information on the extensive behavioural assessment (pp. 26-27):

“Experimental procedure and behavioural assessment

The overall study comprised a 3-hour behavioral testing session including reading, spelling, and general cognitive testing, and two sessions that combined functional neuroimaging (fMRI) and transcranial magnetic stimulation. Please note that to answer the present questions, only data from the ineffective sham (placebo) stimulation session entered the here reported analyses.

During the behavioural session, we did not only assess reading and spelling, but also ruled out that subjects had low nonverbal IQ and comorbidities that could impact our results. The first subtest of the *Culture Fair Test* (CFT 20-R) ⁶⁰ was administered to test participants’ nonverbal intelligence. For all tests, points were added up to calculate raw scores which were then converted into standardized scores. Since no subject had a nonverbal IQ below 85, nobody had to be excluded from further participation. Furthermore, we administered two tests to rule out arithmetic and attention deficits: (1) the *Eggenberger Rechentest* (ERT JE) ⁶¹, a standardised arithmetic test consisting of calculations, conversions and applied mathematics; (2) the *Continuous Attention Performance Test* (CPT – computerized version) ⁶², measuring selective and continuous attention.

We assessed verbal and visuo-spatial working memory to explore potential differences between the two groups. Verbal working memory was assessed with digit span forward and digit span backward from the *WAIS-VI* ⁶³. Subjects had two attempts to repeat the same number of digits. If both were incorrect, the test was terminated. A total of 14 (raw) points could be achieved. Additionally, nonword span was assessed with the Mottier test ⁶⁴. The test was terminated when subjects could repeat < 50% of the nonword syllables. For visual spatial working memory, we employed a computerized version of the Corsi block-tapping test ⁶⁵. Subjects were presented with nine unsystematically arranged black squares on a grey computer display, in which, in short succession, a sequence of squares would light up with increasing length. In the forward and backward condition, the task is to left click on the squares in the same order as presented or in backward order, respectively. The test was stopped if two trials of the same length failed.

In terms of reading, we assessed text, word and pseudoword reading. Silent text reading was assessed with the LGVT 5–12+ ⁶⁶, providing speed (number of words read), accuracy (ratio of filled gaps and correct items) and comprehension scores (number of correctly inserted words). Participants had six minutes to read as far as they could and fill in missing gaps in the text using one of three options. Word and pseudoword reading were assessed with the SLRT-II ⁶⁷. The participants had to correctly read aloud as many words / pseudowords as they could in one minute. The difficulty increased with each column. One raw score was computed for speed and accuracy based on the number of correctly read words / pseudowords in one minute.

Spelling was assessed by the *Rechtschreibungstest (RT)* ⁶⁸, in which participants were asked to fill in missing words of a text, which the examiner read out loud. The percentages of correct answers are based on 68 words.

Phonemic awareness, lexical access and retrieval of phonological representations were tested applying and recording a spoonerism task (German adaption) ⁶⁹, where participants were asked to interchange the initial sounds of the first names and surnames of 12 well-known German personalities of cartoon figures. Response time was measured from the offset of the stimulus to the offset of the response. Finally, to assess reading fluency, we used a rapid automatized naming task with letters, colours and numbers (RAN) ⁷⁰. Naming time was measured and a raw score for items read per second was calculated.”

L571, L602, L607: I cannot identify any systematic search that is conducted when the number of voxels that are included in the PPI and DCM analysis are selected. 25%, 10% or 10% most active voxels are selected in different analyses. Is there a particular reason that these numbers were selected? How reliable are the results when a varying number of voxels were chosen?

This is indeed a good point and we are happy to clarify this. We had tried to keep this constant but it was not possible across the various analyses we performed. For the PPI, we chose functional coupling with the clusters we had found in the univariate analyses, which differed considerably in extent. Therefore, we chose the top 25% of most active voxels, which is a common approach for PPI. We did the same in the DCM based on our univariate findings. Please note that there was a mistake in the methods section – we chose 25%, not 20% (p.32; see below). The only problem we encountered was when we designed the DCM based on the reading network from the literature, where we selected peak coordinates from a meta-analysis on reading processing in left TPC, left IFG and left vOTC (see above also). Here, the clusters of 20mm spheres were huge and covered not only the brain regions we were most interested in, but also adjacent areas. To overcome this, we only included the top 10% of most active voxels, which allowed us to have ROIs in the DCM that were comparable to the DCM with the extended reading network. We hope that this clarifies our choice.

“We performed a two-level analysis using DCM and a Parametric Empirical Bayes (PEB) framework ⁹⁴. At the first level, a full model was specified and estimated for each participant. This full model includes all selected regions and all interactions between these regions. For the first DCM, our full model included the classical reading network from the literature: left TPC, left vOTC, and left IFG. The regions were defined functionally in each individual participant as the top 10% most active voxels for [all trials > rest] within 20 mm spheres around the MNI peak coordinates from the meta-analysis by Martin et al. ⁹⁵: left TPC = -49 -44 21; left IFG = -52 20 18; left vOTC = -42 -68 -22. The second DCM was based on a full model that included our extended reading network, i.e., the four most underactivated clusters (see PPI): left SMG, left vOTC, right vOTC and right cerebellum. These ROIs were defined functionally in each participant as the top 25% of most activate voxels (>80 voxels) for the contrast [all trials > rest].”

L573: Which deconvolution method did the authors use?

Regarding PPI, there is evidence that the results are similar (if not the same) with and without deconvolution (see Di & Biswal, 2017). However, generalized PPI forms the psychophysiological interactions using the deconvolution step as initially described in Gitelman et al. (2003), which we used in our current analysis. As such, gPPI extracts the BOLD signal from chosen ROIs, removes noise and uses this adjusted signal to obtain an estimate for neuronal activity. This canonical hemodynamic response function that is applied in the deconvolution step of PPI seems to be largely constant across subjects and voxels (Gitelman et al., 2003, McLaren et al., 2012).

Di, X., & Biswal, B. B. (2017). Psychophysiological interactions in a visual checkerboard task: Reproducibility, reliability, and the effects of deconvolution. *Frontiers in neuroscience*, 11, 573.

Gitelman, D. R., Penny, W. D., Ashburner, J., & Friston, K. J. (2003). Modeling regional and psychophysiological interactions in fMRI: the importance of hemodynamic deconvolution. *Neuroimage*, 19(1), 200-207.

McLaren, D. G., Ries, M. L., Xu, G., & Johnson, S. C. (2012). A generalized form of context-dependent psychophysiological interactions (gPPI): a comparison to standard approaches. *Neuroimage*, 61(4), 1277-1286.

L617: How many reduced models and comparisons were made? Please also specify which “certain parameters” were switched off.

We now clarify this on page 32 as follows:

“At the second level, DCM parameters of individual participants were entered into a GLM—the PEB model—that decomposed interindividual variability in connection strengths into group effects and random effects⁹⁶. We then compared the full model against 256 reduced models that had parameters “switched off” that did not contribute to the model evidence (i.e., prior mean and variance set to 0)⁹⁴.

Regarding Figure captions, I suggest the authors review all figure captions. In its current form, the figure captions lack the necessary details to understand the figure. For example, a good figure caption would include a brief description of the methods and statistical information necessary to understand the figure without having to refer to the main text.

Thank you for pointing this out. Of course, we do want our figures to be comprehensible and interpretable without having to refer to the main text. Therefore, we have added the requested information in the figure captions. The revised Figure captions read as follows:

Figure 1. fMRI study design and behavioural results. (A) During fMRI, subjects read simple and complex words (red) and simple and complex pseudowords (blue). Mini-blocks always contained 5 items and lasted for 27.5 seconds. Inter-stimulus intervals and rest periods were jittered (2.5 - 4 seconds and 7 - 12 seconds respectively). (B) Linear mixed model results show group-differences in task performance, including speech onsets, reading times and reading accuracy. Significant post-hoc tests are indicated (**p<0.01, ***p<0.001, *p<0.05) and error bars (SE) are shown. Adults with dyslexia read all items slower but needed significantly longer to overtly produce

pseudowords. Moreover, their accuracy was only significantly lower for the pseudoword reading blocks, with worse performance during complex pseudoword reading.

Figure 2. Functional activation results for the direct contrasts between conditions: words vs. pseudowords, simple vs. complex stimuli. For both groups (CG = control group, DYS = dyslexia group), univariate analyses revealed similar activation patterns for the word vs. pseudoword and the simple vs. complex contrasts. Words are shown in red and pseudowords in blue. Simple stimuli are shown in green, complex stimuli in yellow/orange ($p < 0.001$ voxel-level, $p < 0.05$ cluster-wise FWE corrected).

Figure 3. Univariate and multivariate group comparisons for the reading tasks. (A) Univariate analyses show hypoactivation in individuals with dyslexia (DYS) relative to the control group (CG) in several regions in- and outside the classical reading network, such as the left SMG, left and right vOTC and right cerebellum. (B) MVPA revealed additional regions where adults with and without dyslexia showed differences in activation patterns. These additional regions comprised the left insula, the bilateral anterior cingulate, the right precuneus and the left cerebellum (extending to the left vOTC) (all analyses: $p < 0.001$ voxel-level, $p < 0.05$ cluster-wise FWE corrected).

Figure 4. Individual activation profiles in the left vOTC, left SMG, right vOTC, right cerebellum and the bilateral lingual gyrus for all four conditions: simple words, complex words, simple pseudowords and complex pseudowords. Parameter estimates are based on the respective rest contrasts (simple words > rest, complex words > rest, simple pseudowords > rest, complex pseudowords > rest). We found high consistency in activation within groups, suggesting that the reported hypoactivation (see **Figure 4**) is consistent across subjects in the dyslexia group. Additionally, the control group (CG) showed an increase in activation of the left and right vOTC for increasing difficulty, and a pseudoword-specific recruitment of the left SMG. Also, the bilateral lingual gyri showed increased activation for complex stimuli in the CG. The only activity patterns observable in both groups were that complex pseudowords led to increased activity in the right cerebellum.

Figure 5. Differences in task-specific functional coupling between the control group and the dyslexia group. We display PPI results for seeds from the univariate contrast results: (A) left vOTC, (B) left SMG, (C) right vOTC, (D) right cerebellum ($p < 0.001$ voxel-level, $p < 0.05$ cluster-wise FWE corrected). The plots depict the respective seed region (yellow) and areas which revealed significantly more connectivity with the seed region in typical readers relative to individuals with dyslexia during word (red) and pseudoword (blue) reading. We found disruptions of functional connectivity between left SMG and left vOTC, as well as between left SMG and bilateral cerebellar and visual areas. The strongest effects were found for the right cerebellum (see D), where adults with dyslexia showed much lower functional connectivity during word and pseudoword reading to large clusters spread across the two hemispheres. For connectivity between left vOTC and left / right cuneus, we found stronger coupling for individuals with dyslexia than controls (see barplots in A and C). This significant effect, however, did not stem from stronger connectivity between these regions in adults with dyslexia, but from the difference in recruitment for word and pseudoword reading.

Figure 6. Differences in effective connectivity between adults with and without dyslexia in the classical (A) and the extended reading network (B). We display the

best model for each group (posterior probability > 95), as well as comparison DCMs across groups. Based on our DCMs, we report differences in connectivity within the classical reading network (A; including left IFG, left TPC and left vOTC). In the control group (CG), words decreased the connectivity via the dorsal route (left TPC to other areas), whereas pseudowords partially relied upon this route. In adults with dyslexia (DYS), in contrast, both words and pseudowords recruited connectivity to and from the left TPC. A direct comparison between both DCMs shows significantly stronger recruitment of the dorsal route during word reading in dyslexia, and stronger left TPC to IFG coupling in typical readers. DCMs of the extended reading network (B; including left SMG, left vOTC, right vOTC and right cerebellum) revealed almost no modulation by stimulus type and strong control of the left vOTC over the left SMG and the right vOTC. In adults with dyslexia, both words and pseudowords differentially recruited the various pathways between regions of the extended reading network.

Figure 7. Effective connectivity-behaviour links. We display Bayesian correlations ($BF_{10} > 7$) between measures of effective connectivity within the classical (A) and extended (hypoactive) (B) reading network and task performance in the scanner in controls (CG) and adults with dyslexia (DYS). Across all correlations, higher connectivity in the dyslexia group was linked to worse reading performance. In other words, individuals with strongest coupling showed weakest performance. While left SMG to left vOTC coupling seemed to be crucial for complex pseudoword reading in dyslexia, left TPC to IFG and left vOTC coupling were more important for word reading.

Figure 8. Schematic illustration of the dorsal route in dyslexia. Based on our DCM findings for the classical and extended reading network, we suggest that both words and pseudowords are at least partially processed via a dorsal route, but the engagement of TPC subregions might be crucial. Effective connectivity analyses suggest that the left vOTC to left pSTG (TPC) connection was modulated by words, whereas the connection to the left SMG was modulated by pseudowords in dyslexia. Our behavioural correlations support their implication for word and pseudoword reading.

The authors mainly show subject-averaged results, but individual subject variability in language processing needs to be considered. The authors do acknowledge “large individual variability in the dyslexia group” (L555-556) but do not show individual subject results. It would be important to show the evidence that results replicate in each individual subject (Fedorenko 2012).

While we only suggested large individual variability in behavioural performance in the dyslexia group, we do see your point. Therefore, we have created plots to show that the claims we make based on our fMRI data is consistent across subjects and show these results in Figure 4.

ACTIVATION MAGNITUDES IN READING-RELATED BRAIN REGIONS

Figure 4. Individual activation profiles in the left vOTC, left SMG, right vOTC, right cerebellum and the bilateral lingual gyrus for all four conditions: simple words, complex words, simple pseudowords and complex pseudowords. Parameter estimates are based on the respective rest contrasts (simple words > rest, complex words > rest, simple pseudowords > rest, complex pseudowords > rest). We found high consistency in activation within groups, suggesting that the reported hypoactivation (see **Figure 4**) is consistent across subjects in the dyslexia group. Additionally, the control group (CG) showed an increase in activation of the left and right vOTC for increasing difficulty, and a pseudoword-specific recruitment of the left SMG. Also, the bilateral lingual gyri showed increased activation for complex stimuli in the CG. The only activity patterns observable in both groups were that complex pseudowords led to increased activity in the right cerebellum.

3) L83: I find the statement “neural marker” (also in the title) rather strong given the complexity of the processes and their yet unresolved nature.

We agree that it is a strong claim and have considered calling it a ‘potential’ neural marker. However, making a weaker claim seems like we are not really sure that connectivity does play a role. Given that we do find these strong effects in all our connectivity analyses, we believe that we have enough evidence to support this claim.

4) L 484-486: “Subjects were instructed to overtly read stimuli as fast and accurately as

possible while avoiding head movements.”

Head motion during overt reading can strongly affect the SNR. How much motion was visible in each scan session? How was head movement controlled / “avoided”? Did the authors assess how SNR is affected with and without overt reading?

Also, was a noise-canceling microphone used to process the in-scanner overt responses?

Yes, we used a noise-cancelling microphone to record the overt reading responses. To control for movement artifacts, we included the motion parameters from realignment into the subject-level general linear model (GLM). To further improve motion confound regression, we also added the motion parameters’ temporal derivatives, quadratic terms, and temporal derivatives of the quadratic terms. Moreover, we performed “motion scrubbing” to remove individual time points with strong volume-to-volume movement from the analysis. To this end, we computed framewise displacement (FD) as a measure of excessive volume-to-volume movement, and added individual regressors for volumes that exceeded a threshold of $FD > 0.9$, as proposed by Siegel et al. (2014) for task-based fMRI data. This has been more extensively described now in the Data analysis section (see p. 29):

“We performed a whole-brain random-effects group analysis based on the general linear model (GLM), using the two-level approach in *SPM12*. First, individual participant data were modeled separately. The participant-level GLM included regressors for the four experimental conditions (simple words, complex words, simple pseudowords, complex pseudowords), modelling trials as box car functions (2.5 seconds duration) convolved with the canonical hemodynamic response function (HRF). Only correct trials with given answers (incorrect trials were trials without answer or with completely wrong words / pseudowords) were analyzed, error trials were modeled in a separate regressor-of-no-interest. To control for movement artifacts, we included the motion parameters from realignment into the subject-level GLM. To further improve motion confound regression, we also added the motion parameters’ temporal derivatives, quadratic terms, and temporal derivatives of the quadratic terms. Therefore, nuisance regressors included 24 motion regressors (the 6 base motion parameters + 6 temporal derivatives of the motion parameters + 12 quadratic terms of the motion parameters and their temporal derivatives)^{78,79}. Moreover, we performed “motion scrubbing” to remove individual time points with strong volume-to-volume movement from the analysis⁸⁰. To this end, we computed framewise displacement (FD) as a measure of excessive volume-to-volume movement and added individual regressors for volumes that exceeded a threshold of $FD > 0.9$, as proposed for task-based fMRI data⁸¹. Finally, we included the top 10 aCompCor regressors explaining the most variance in physiological noise⁷⁷. The data were subjected to an AR(1) auto-correlation model to account for temporal auto-correlations, and high-pass filtered (cutoff 128s) to remove low-frequency noise.”

5) L570: *I suggest citing the original papers when describing PPI:*

Friston KJ, Buechel C, Fink GR, Morris J, Rolls E, Dolan RJ. Psychophysiological and modulatory interactions in neuroimaging. Neuroimage. 1997; 6:218–229.

Gitelman DR, Penny WD, Ashburner J, Friston KJ. Modeling regional and psychophysiological interactions in fMRI: the importance of hemodynamic deconvolution. Neuroimage. 2003; 19:200207.

Thanks for suggesting this. The second paper had been cited in the PPI section already, but we added it in the first sentence, where we describe the main mechanism of PPI. We have also added the Friston et al. (1997) original paper in that same sentence.

Minor comments

1) L10: *“Our results reveal disrupted functional coupling between hypoactive reading regions and large clusters across the brain, with the right cerebellum showing the largest disruption”*: Please specify *“large clusters across the brain”*.

Due to the word limit for the abstract, we were not able to specify this in the abstract but have eliminated this vague statement. The abstract now states:

“We find disrupted functional coupling between hypoactive reading regions, especially between the left temporo-parietal and occipito-temporal cortex, and an extensive functional disruption of the right cerebellum in adults with dyslexia.”

2) L22-23: *“Stronger connectivity within networks is linked to worse reading performance”*
Specify to which subject group this finding refers to.

Done.

“Stronger connectivity within networks is linked to worse reading performance in dyslexia.”

3) L 84, *Results*: *To make the results section more comprehensible, I suggest the authors include one paragraph at the beginning of the section that describes the specific experimental design referring to Figure 1, the tasks, and study goals. This will help to better/faster understand L96-102.*

We agree that this makes the results section more accessible and have thus reiterated the goals and the specific task design at the beginning of the results section:

“The present study investigated reading and spellings skills and the processing of simple and complex words and pseudowords during functional neuroimaging in typical readers and adults with dyslexia. We aimed to (1) detect potential differences in terms of functional activation during reading, (2) investigate functional connectivity between reading-relevant regions in typical and atypical readers, and (3) explore effective connectivity, i.e., directed coupling, within the reading network. Moreover, we investigated whether differences in functional activation and coupling were linked to behavioural performance in and outside the MRI.”

4) L87-90: *Table 1: A short description of the tasks included as supplemental material would be helpful.*

We agree that it does make sense to go more into detail here. Therefore, we have extended this section in the methods (instead of the supplements) (pp. 26-27):

“Experimental procedure and behavioural assessment

The overall study comprised a 3-hour behavioral testing session including reading, spelling, and general cognitive testing, and two sessions that combined functional neuroimaging (fMRI) and transcranial magnetic stimulation. Please note that to answer the present questions, only data from the ineffective sham (placebo) stimulation session entered the here reported analyses.

During the behavioural session, we did not only assess reading and spelling, but also ruled out that subjects had low nonverbal IQ and comorbidities that could impact our results. The first subtest of the *Culture Fair Test* (CFT 20-R)⁶⁰ was administered to test participants' nonverbal intelligence. For all tests, points were added up to calculate raw scores which were then converted into standardized scores. Since no subject had a nonverbal IQ below 85, nobody had to be excluded from further participation. Furthermore, we administered two tests to rule out arithmetic and attention deficits: (1) the *Eggenberger Rechentest* (ERT JE)⁶¹, a standardised arithmetic test consisting of calculations, conversions and applied mathematics; (2) the *Continuous Attention Performance Test* (CPT – computerized version)⁶², measuring selective and continuous attention.

We assessed verbal and visuo-spatial working memory to explore potential differences between the two groups. Verbal working memory was assessed with digit span forward and digit span backward from the *WAIS-VI*⁶³. Subjects had two attempts to repeat the same number of digits. If both were incorrect, the test was terminated. A total of 14 (raw) points could be achieved. Additionally, nonword span was assessed with the Mottier test⁶⁴. The test was terminated when subjects could repeat < 50% of the nonword syllables. For visual spatial working memory, we employed a computerized version of the Corsi block-tapping test⁶⁵. Subjects were presented with nine unsystematically arranged black squares on a grey computer display, in which, in short succession, a sequence of squares would light up with increasing length. In the forward and backward condition, the task is to left click on the squares in the same order as presented or in backward order, respectively. The test was stopped if two trials of the same length failed.

In terms of reading, we assessed text, word and pseudoword reading. Silent text reading was assessed with the LGVT 5–12+⁶⁶, providing speed (number of words read), accuracy (ratio of filled gaps and correct items) and comprehension scores (number of correctly inserted words). Participants had six minutes to read as far as they could and fill in missing gaps in the text using one of three options. Word and pseudoword reading were assessed with the SLRT-II⁶⁷. The participants had to correctly read aloud as many words / pseudowords as they could in one minute. The difficulty increased with each column. One raw score was computed for speed and accuracy based on the number of correctly read words / pseudowords in one minute. Spelling was assessed by the *Rechtschreibungstest* (RT)⁶⁸, in which participants were asked to fill in missing words of a text, which the examiner read out loud. The percentages of correct answers are based on 68 words.

Phonemic awareness, lexical access and retrieval of phonological representations were tested applying and recording a spoonerism task (German adaption)⁶⁹, where participants were asked to interchange the initial sounds of the first names and surnames of 12 well-known German personalities of cartoon figures. Response time was measured from the offset of the stimulus to the offset of the response. Finally, to assess reading fluency, we used a rapid automatized naming task with letters, colours

and numbers (RAN) ⁷⁰. Naming time was measured and a raw score for items read per second was calculated.”

5) L127, Figure2: The colormap used to visualize “Simple vs. Complex” conditions is very hard to distinguish. I suggest changing this to a different colormap with stronger color contrast.

We chose the colours green and yellow and remade the figure. We hope the contrast is now easier to see.

6) L179: Figure 4A and 4C: Why are the bar plots only highlighted for the connectivity between L vOTC and L/R cuneus?

Only connectivity between two areas showed higher functional coupling for the dyslexia group as compared to the control group. To disentangle this effect, we provided bar plots. We have specified this in the figure description. Please note that this is now Figure 5.

Figure 5. Differences in task-specific functional coupling between the control group and the dyslexia group. We display PPI results for seeds from the univariate contrast results: (A) left vOTC, (B) left SMG, (C) right vOTC, (D) right cerebellum ($p < 0.001$ voxel-level, $p < 0.05$ cluster-wise FWE corrected). The plots depict the respective seed region (yellow) and areas which revealed significantly more connectivity with the seed region in typical readers relative to individuals with dyslexia during word (red) and pseudoword (blue) reading. We found disruptions of functional

connectivity between left SMG and left vOTC, as well as between left SMG and bilateral cerebellar and visual areas. The strongest effects were found for the right cerebellum (see D), where adults with dyslexia showed much lower functional connectivity during word and pseudoword reading to large clusters spread across the two hemispheres. For connectivity between left vOTC and left / right cuneus, we found stronger coupling for individuals with dyslexia than controls (see barplots in A and C). This significant effect, however, did not stem from stronger connectivity between these regions in adults with dyslexia, but from the difference in recruitment for word and pseudoword reading.

7) L326: *Typo in “showws reduced functional [...]”*

Thank you - changed.

8) L377: *Typo “we” in “Interestingly, we the direct group comparison also showed increased coupling from [...].”*

Thank you – changed.

9) L458: *Describe which specific behavioral tests were used.*

We have added this specific information in the methods section on Experimental procedure and behavioural assessment – please see our answer to your comment above!

10) L501-506: *Please cite the corresponding manuscripts for ANT, Freesurfer, AFNI, and SPM.*

We have added the corresponding references to the methods section.

11) L518: *It would be useful to list the four experimental conditions here. In Methods and Figure 1, these conditions are mentioned but not explicitly. I would also suggest making the four experimental conditions in L491-492 in Methods explicit.*

We have added details on the tasks and conditions. In the methods section, the task is now described as follows:

“We chose a mini-block design with a reading task comprising four conditions – simple words, complex words, simple pseudowords and complex pseudowords (see Stimuli). During fMRI, stimuli were presented for 2.5 seconds and grouped in mini blocks of 5 stimuli (**Figure 1A**). Subjects’ task was to read the items they saw aloud. We jittered the inter-stimulus-interval (2.5 - 4 seconds) and the rest time, i.e., the interval between mini blocks (7 - 12 seconds). Each mini-block lasted for 27.5 seconds resulting in a total scanning time of 25 minutes. Subjects were instructed to overtly read stimuli as fast and accurately as possible while avoiding head movements. Subjects’ in-scanner responses were recorded and manually preprocessed using *Audacity*. Speech on- and offsets were determined using *Praat* by four

independent raters and 50% of all responses were rated by two of these four raters. We calculated an interrater reliability of >0.85 , suggesting high reliability.

Stimuli

We used an event-related mini-block design in which subjects overtly read 100 words (50 simple, 50 complex) and 100 pseudowords (50 simple, 50 complex). The simple, 2-syllable (4-6 letters) words were those provided by Schuster et al. (2015). Complex words comprised the 100 most frequent 4-syllabic German words (10-14 letters) from the *dlex* database (<http://www.dlexdb.de/>). We excluded compound words and plurals but included some 3-syllabic words and plurals due to a lack of 4-syllabic words that were neither compounds nor plurals. Pseudowords were generated using the *Wuggy* software (<http://crr.ugent.be/programs-data/wuggy>) based on the simple and complex word lists. Thus, all pseudowords followed the phonotactic rules of German. Stimuli were randomized in blocks and blocks were randomized within the run. Each item was only shown once. We excluded pseudowords with a < 2 letters difference to real German words.”

12) L540: Was there a specific reason why a “leave-two-participants-out” procedure was chosen over the more common “leave-one-participant-out” procedure?

We chose to leave two participants (one participant per group) out to test whether the classifier indeed learned to correctly classify both people with dyslexia and controls, without a bias towards one particular group. Note that a classifier which is completely biased towards people with dyslexia (i.e., a classifier that classifies a person as dyslexic in 100% of cases) would perform perfectly in a leave-one-participant-out procedure where the left-out participant has dyslexia. We have added the following sentence to the Methods section (p. 30):

“We performed leave-two-*participants*-out cross validation (CV), training on the activation patterns from $n-1$ participants per group and testing on the left-out two participants (i.e., 1 from each group; yielding 26 CV-folds). This procedure was chosen to test whether the classifier indeed learned to correctly classify people with dyslexia and controls without a bias towards either group.”

Reviewers' comments:

Reviewer #1 (Remarks to the Author):

The authors have adequately responded to comments on the original version of the MS.

Minor issues

Fig S1 involves only PW>rest comparison data

Reviewer #3 (Remarks to the Author):

I appreciate the authors' efforts in revising the Figures and Methods section, as well as incorporating the activation patterns of individual subjects into their manuscript. These changes have enhanced the manuscript's clarity and comprehensibility. However, I have some remaining comments.

(1) Voxel selection

Regarding voxel selection for the PPI and DCM analysis. First, in their rebuttal, the authors mention that choosing the top 25% of most active voxels "is a common approach for PPI" but have not included a reference. This could be helpful to understand the nature of this number. It is still unclear to me where this number is coming from. Has this been systematically tested before? How reliable are the results when a varying number of voxels were chosen? This could be done by a hyperparameter search. For example by defining an interval of percentages, systematically varying this percentage by a small step, and then selecting the most robust percentage.

(2) Motion correction

The authors used the motion censoring method described in Siegel et al. Could the authors show the motion traces of subjects and describe how many data points were censored per subject?

I have not applied motion censoring but have been following the concerns about the usage of the method in the literature. See for example Power et al. 2015, Caballero-Gaudes and Reynolds, 2017, Eklund et al. 2020. Hence, I am a little concerned that for example some of the analysis steps (e.g., autocorrelation) in combination with the censoring maybe interfere with the connectivity results. Could the authors elaborate on the potential effects of motion scrubbing used in their connectivity results and ideally present necessary controls that they may have done?

References:

Caballero-Gaudes, C., & Reynolds, R. C. (2017). Methods for cleaning the BOLD fMRI signal. *Neuroimage*, 154, 128-149.

Power, J. D., Schlaggar, B. L., & Petersen, S. E. (2015). Recent progress and outstanding issues in motion correction in resting state fMRI. *Neuroimage*, 105, 536-551.

Eklund, A., Nichols, T. E., Afyouni, S., & Craddock, C. (2020). How does group differences in motion scrubbing affect false positives in functional connectivity studies?. *BioRxiv*, 2020-02.

(3) L710: Author's mentioned in the rebuttal that they used a noise-canceling microphone to record the overt responses. This should be described. I assume a specific hardware (OptoAcoustics or others) must have been used, so this information is missing.

(4) L711: Please include citations for the used software Audacity and Praat.

(5) L751: Period is missing in "To control for movement artifacts, we included the motion parameters

from realignment into the subject-level”.

(6) L771-773: “Studies using simulations suggest that multivariate pattern analysis (MVPA) is sensitive to the magnitude of variability found on the voxel level (i.e., spatial variation), and thus more sensitive to variability on the subject-level, which is a major shortcoming of univariate analyses.”

There is something not matching in this argument. The cited article by Davis et al. 2015 shows that MVPA maybe more powerful than univariate voxelwise tests because MVPA (1) makes use of voxel-level variability within subjects, (2) is insensitive to/discards subject-level variability in mean activation. The authors claim that MVPA is more sensitive to variability on the subject-level.

(7) L791: Please include citations for the used software JASP and R.

How to cite R: <https://intro2r.com/citing-r.html>

How to cite JASP: <https://jasp-stats.org/faq/how-do-i-cite-jasp/>

Dear Reviewers,

Please find our detailed responses to your feedback (in *italics*) under the corresponding comment / paragraph below.

Reviewer 1:

The authors have adequately responded to comments on the original version of the MS.

Minor issues

Fig S1 involves only PW>rest comparison data

Thank you for your positive comment. We are happy that all of your concerns and comments were sufficiently addressed in the revised version of the manuscript.

We do not really understand your comment on **Figure S1**. We copied **Figure S1** again to show that it does not only include the PW vs rest comparison data. Please note the differences in activation between the four conditions. Maybe there was a misunderstanding or something was wrong in our reviewer comments. Please note that the differences in recruitment of words vs. pseudowords stem from large patterns of deactivation, which are not visible in the condition vs. rest contrasts but only in the direct comparison. Maybe you wanted to point out that we did not provide figures for rest > conditions? We are happy to provide these and have attached them to the supplements (see **Figures S3 and S4** below).

Figure S1. Functional activation results for all four conditions (simple words, complex words, simple pseudowords, complex pseudowords) vs. rest in the control group ($p < 0.001$ voxel-level, $p < 0.05$ cluster-wise FWE corrected).

Figure S3. Functional activation results for rest > all four conditions (simple words, complex words, simple pseudowords, complex pseudowords) in the control group ($p < 0.001$ voxel-level, $p < 0.05$ cluster-wise FWE corrected).

Figure S4. Functional activation results for rest > all four conditions (simple words, complex words, simple pseudowords, complex pseudowords) in the dyslexia group ($p < 0.001$ voxel-level, $p < 0.05$ cluster-wise FWE corrected).

Reviewer 3:

I appreciate the authors' efforts in revising the Figures and Methods section, as well as incorporating the activation patterns of individual subjects into their manuscript. These changes have enhanced the manuscript's clarity and comprehensibility. However, I have some remaining comments.

Thank you for your positive feedback. We are happy that some of your concerns have already been adequately addressed. Please see our responses to the two main remaining issues below.

(1) Voxel selection

Regarding voxel selection for the PPI and DCM analysis. First, in their rebuttal, the authors mention that choosing the top 25% of most active voxels "is a common approach for PPI" but have not included a reference. This could be helpful to understand the nature of this number. It is still unclear to me where this number is coming from. Has this been systematically tested before? How reliable are the results when a varying number of voxels were chosen? This could be done by a hyperparameter search. For example, by defining an interval of percentages, systematically varying this percentage by a small step, and then selecting the most robust percentage.

We agree that it was misleading to have called our choice of including the 25% active voxels for each ROI a 'common approach for PPI'. Most studies of the past years that have applied PPI chose their ROIs based on a single peak voxel and then drew a random-sized sphere (mostly between 4 and 8mm) around this peak coordinate, assuming that it captured activation within that region in each subject (e.g., La et al., 2016; Stroh et al., 2019; Trimmel et al., 2018). However, activation patterns can differ substantially between individuals (Fedorenko & Kanwisher, 2011). Especially in the case of individuals with atypical activation patterns (e.g. people with dyslexia), we believe that it is more meaningful to select the most active voxels in each individual subject. It has been shown that this approach yields higher sensitivity and functional resolution (i.e., the ability to separate adjacent but functionally distinct regions) than the classical approach of defining ROIs based on the same location in standard space (Nieto-Castañón & Fedorenko, 2012). Initially, we had aimed to use an ROI size selection of 10% of the most active voxels as suggested by Fedorenko and colleagues (Basilakos et al., 2018; Fedorenko et al., 2012) and also applied in studies of our own group (Kuhnke et al., 2021). The problem was that this ROI size definition led to a very small number of voxels in each cluster since we chose the ROIs based on the hypoactive clusters from the whole-brain univariate analyses. Using only 10% most active voxels yielded 64 voxels for left vOTC, 39 voxels for left SMG, 12 voxels for right vOTC and 9 voxels for the right cerebellum. Clusters of 9-12 voxels are much smaller than a typical seed ROI for PPI (e.g., 4 to 8mm radius spheres). Therefore, we increased the percentage of included active voxels to achieve clusters of at least 20 voxels in each ROI, resulting in the top 25% of active voxels within that region. Although this is still smaller than in many studies (e.g., in the meta-analysis of Smith et al., 2016, seed clusters ranged from 95a to 326 voxels), it is definitely superior to the 10% selection.

However, we fully agree that the choice of 25% most active voxels is arbitrary. Therefore, we have now re-run all of our PPI analyses with seed ROIs based on the 10% most active voxels in each subject. Importantly, the results are highly similar to the results of our original analyses based on the 25% most active voxels, with minor differences (see figures below; references can be found at the end of the document).

Results of the 25% most active voxel ROI selection (as reported in Manuscript):

Results of 10% most active voxels ROI selection:

(1) **Seed in left SMG:** CG > DYS (red: words; blue: pseudowords)

(2) **Seed in left vOTC:** CG > DYS (red: words; blue: pseudowords)

(3) **Seed in right vOTC:** CG > DYS (red: words; blue: pseudowords)

(4) **Seed in right cerebellum:** CG > DYS (red: words; blue: pseudowords); as in previous analysis, large clusters spreading over both hemispheres. For illustration purposes, we plotted them with Brainnetviewer.

(2) Motion correction

The authors used the motion censoring method described in Siegel et al. Could the authors show the motion traces of subjects and describe how many data points were censored per subject?

I have not applied motion censoring but have been following the concerns about the usage of the method in the literature. See for example Power et al. 2015, Caballero-Gaudes and Reynolds, 2017, Eklund et al. 2020. Hence, I am a little concerned that for example some of the analysis steps (e.g., autocorrelation) in combination with the censoring maybe interfere with the connectivity results. Could the authors elaborate on the potential effects of motion scrubbing used in their connectivity results and ideally present necessary controls that they may have done?

Thank you for pointing this out. We wish to make clear that we did not apply a method where the time series per se is changed (e.g., by deleting entire volumes during which subjects show too much movement). We only added single-volume regressors for each time point with high volume-to-volume movement (framewise displacement > 0.9) to our subject-level GLMs. The studies you cite are largely based on resting-state data (Power et al., Eklund et al.) and not task-based fMRI data. We believe that in resting-state studies, artifacts might affect interpretation much more as compared to task-based fMRI studies like ours, where the influence of the task dominates the BOLD signal (Siegel et al., 2014). Therefore, we believe that our motion censoring approach is well-suited for task-based studies. To show that we have paid sufficient attention to monitoring potential movement artifacts, we have prepared a table of the framewise displacement values for all subject. We are happy to add this to the supplement if you feel that this is important for the present study.

Subject	group	num_outliers	max_FD	mean_FD	std_FD
Sub-1	DYS	6	1.2959	0.2581	0.1651
Sub-2	DYS	2	0.9470	0.2917	0.1397
Sub-3	DYS	0	0.6794	0.1942	0.0995
Sub-4	DYS	39	5.9748	0.4150	0.3661
Sub-5	DYS	0	0.8007	0.2588	0.1209
Sub-6	DYS	0	0.7446	0.2026	0.1024
Sub-7	DYS	30	2.0691	0.4069	0.2718
Sub-8	DYS	59	1.7479	0.4661	0.2676
Sub-9	DYS	1	1.0999	0.1714	0.0939
Sub-10	DYS	0	0.6410	0.2076	0.1055

Sub-11	DYS	0	0.4630	0.1667	0.0693
Sub-12	DYS	0	0.4167	0.1490	0.0643
Sub-13	DYS	1	0.9234	0.2681	0.1265
Sub-14	DYS	1	0.9278	0.1920	0.1126
Sub-15	DYS	0	0.4910	0.1866	0.0849
Sub-16	DYS	1	1.0885	0.2065	0.1148
Sub-17	DYS	5	1.3238	0.3810	0.1902
Sub-18	DYS	21	6.1666	0.3328	0.3573
Sub-19	DYS	1	1.1901	0.2220	0.1190
Sub-20	DYS	15	2.9056	0.2339	0.2333
Sub-21	DYS	0	0.4390	0.1884	0.0856
Sub-22	DYS	1	0.9856	0.1970	0.1141
Sub-23	DYS	4	1.3790	0.2834	0.1580
Sub-24	DYS	6	2.6000	0.2080	0.1750
Sub-25	DYS	0	0.8364	0.1641	0.0882
Sub-26	DYS	0	0.8811	0.2253	0.1231
sub-1	CG	0	0.7690	0.2669	0.1273
sub-2	CG	11	1.6991	0.3850	0.1878
sub-3	CG	14	1.4774	0.3425	0.2075
sub-4	CG	2	0.9516	0.2716	0.1602
sub-5	CG	0	0.5382	0.1706	0.0761
sub-6	CG	19	1.5951	0.3422	0.2124
sub-7	CG	0	0.5043	0.2006	0.0832
sub-8	CG	0	0.4668	0.1114	0.0553
sub-9	CG	0	0.8473	0.1644	0.1105
sub-10	CG	0	0.7335	0.2163	0.0978
sub-11	CG	0	0.8504	0.2956	0.1499
sub-12	CG	6	1.8085	0.2695	0.1742
sub-13	CG	0	0.8594	0.2215	0.1325
sub-14	CG	0	0.6814	0.1891	0.0878
sub-15	CG	0	0.8589	0.1680	0.1033
sub-16	CG	0	0.6940	0.1516	0.0862
sub-17	CG	3	1.2210	0.3112	0.1507
sub-18	CG	0	0.8289	0.2679	0.1311
sub-19	CG	0	0.4869	0.1497	0.0793
sub-20	CG	2	2.3739	0.1669	0.1447
sub-21	CG	78	8.3539	0.5476	0.8127
sub-22	CG	0	0.4749	0.1285	0.0703
sub-23	CG	6	1.2003	0.2565	0.1602
sub-24	CG	0	0.5380	0.1234	0.0661
sub-25	CG	1	0.9580	0.2284	0.1209
sub-26	CG	0	0.7583	0.1890	0.1023
sub-27	CG	8	1.4499	0.2650	0.1870
sub-28	CG	0	0.7833	0.2106	0.1129

References:

Caballero-Gaudes, C., & Reynolds, R. C. (2017). Methods for cleaning the BOLD fMRI signal. Neuroimage, 154, 128-149.

Power, J. D., Schlaggar, B. L., & Petersen, S. E. (2015). Recent progress and outstanding issues in motion correction in resting state fMRI. Neuroimage, 105, 536-551.

Eklund, A., Nichols, T. E., Afyouni, S., & Craddock, C. (2020). How does group differences in motion scrubbing affect false positives in functional connectivity studies?. BioRxiv, 2020-02.

(3) L710: Author's mentioned in the rebuttal that they used a noise-canceling microphone to record the overt responses. This should be described. I assume a specific hardware (OptoAcoustics or others) must have been used, so this information is missing.

We have added this information in the methods section:

“To record subjects’ responses during MRI, an Optoacoustics FOMRI-III dual channel microphone system was used.”

(4) L711: Please include citations for the used software Audacity and Praat.

Done – thank you four pointing this out!

(5) L751: Period is missing in “To control for movement artifacts, we included the motion parameters from realignment into the subject-level”.

Thank you, this has been changed.

(6) L771-773: “Studies using simulations suggest that multivariate pattern analysis (MVPA) is sensitive to the magnitude of variability found on the voxel level (i.e., spatial variation), and thus more sensitive to variability on the subject-level, which is a major shortcoming of univariate analyses.”

There is something not matching in this argument. The cited article by Davis et al. 2015 shows that MVPA maybe more powerful than univariate voxelwise tests because MVPA (1) makes use of voxel-level variability within subjects, (2) is insensitive to/discards subject-level variability in mean activation. The authors claim that MVPA is more sensitive to variability on the subject-level.

That was indeed a mistake from our side. The sentence has been changed accordingly:

“Studies using simulations suggest that multivariate pattern analysis (MVPA) is sensitive to the magnitude of variability found on the voxel level (i.e., spatial variation), which is a major shortcoming of univariate analyses.”

Univariate analyses are only sensitive to absolute changes in the mean activation magnitude of brain regions (Raizada & Kriegeskorte, 2010). However, neural representations of mental contents are generally assumed to be encoded in “population codes”—*relative* differences in activation patterns distributed across multiple voxels (Connolly et al., 2011; Haxby et al., 2014; Ritchie et al., 2019)(Connolly et al., 2012; Haxby et al., 2014; Ritchie et al., 2019). Whereas univariate analyses are insensitive to such fine-grained activity pattern differences, population codes can be studied using multivariate pattern analyses (MVPA) of functional neuroimaging data (Kuhnke et al., 2023; Mur et al., 2009).

(7) L791: Please include citations for the used software JASP and R.

How to cite R: <https://intro2r.com/citing-r.html>

How to cite JASP: <https://jasp-stats.org/faq/how-do-i-cite-jasp/>

Done – thank you four pointing this out!

- Basilakos, A., Smith, K. G., Fillmore, P., Fridriksson, J., & Fedorenko, E. (2018). Functional Characterization of the Human Speech Articulation Network. *Cerebral Cortex (New York, NY)*, 28(5), 1816–1830. <https://doi.org/10.1093/cercor/bhx100>
- Connolly, A. C., Gobbini, M. I., & Haxby, J. V. (2011). Three Virtues of Similarity-Based Multivariate Pattern Analysis: An Example from the Human Object Vision Pathway. In N. Kriegeskorte & G. Kreiman (Eds.), *Visual Population Codes* (pp. 335–356). The MIT Press. <https://doi.org/10.7551/mitpress/8404.003.0016>
- Fedorenko, E., & Kanwisher, N. (2011). Functionally Localizing Language-Sensitive Regions in Individual Subjects With fMRI: A Reply to Grodzinsky's Critique of Fedorenko and Kanwisher (2009). *Language and Linguistics Compass*, 5(2), 78–94. <https://doi.org/10.1111/j.1749-818X.2010.00264.x>
- Fedorenko, E., McDermott, J. H., Norman-Haignere, S., & Kanwisher, N. (2012). Sensitivity to musical structure in the human brain. *Journal of Neurophysiology*, 108(12), 3289–3300. <https://doi.org/10.1152/jn.00209.2012>
- Haxby, J. V., Connolly, A. C., & Guntupalli, J. S. (2014). Decoding Neural Representational Spaces Using Multivariate Pattern Analysis. *Annual Review of Neuroscience*, 37(1), 435–456. <https://doi.org/10.1146/annurev-neuro-062012-170325>
- Kuhnke, P., Kiefer, M., & Hartwigsen, G. (2021). Task-Dependent Functional and Effective Connectivity during Conceptual Processing. *Cerebral Cortex*, 31(7), 3475–3493. <https://doi.org/10.1093/cercor/bhab026>
- Kuhnke, P., Kiefer, M., & Hartwigsen, G. (2023). Conceptual representations in the default, control and attention networks are task-dependent and cross-modal. *Brain and Language*, 244, 105313. <https://doi.org/10.1016/j.bandl.2023.105313>
- La, C., Garcia-Ramos, C., Nair, V. A., Meier, T. B., Farrar-Edwards, D., Birn, R., Meyerand, M. E., & Prabhakaran, V. (2016). Age-Related Changes in BOLD Activation Pattern in Phonemic Fluency Paradigm: An Investigation of Activation, Functional Connectivity and Psychophysiological Interactions. *Frontiers in Aging Neuroscience*, 8. <https://www.frontiersin.org/articles/10.3389/fnagi.2016.00110>
- Mur, M., Bandettini, P. A., & Kriegeskorte, N. (2009). Revealing representational content with pattern-information fMRI—an introductory guide. *Social Cognitive and Affective Neuroscience*, 4(1), 101–109. <https://doi.org/10.1093/scan/nsn044>
- Nieto-Castañón, A., & Fedorenko, E. (2012). Subject-specific functional localizers increase sensitivity and functional resolution of multi-subject analyses. *NeuroImage*, 63(3), 1646–1669. <https://doi.org/10.1016/j.neuroimage.2012.06.065>
- Raizada, R. D. S., & Kriegeskorte, N. (2010). Pattern-information fMRI: New questions which it opens up and challenges which face it. *International Journal of Imaging Systems and Technology*, 20(1), 31–41. <https://doi.org/10.1002/ima.20225>
- Ritchie, J. B., Kaplan, D. M., & Klein, C. (2019). Decoding the Brain: Neural Representation and the Limits of Multivariate Pattern Analysis in Cognitive Neuroscience. *The British Journal for the Philosophy of Science*, 70(2), 581–607. <https://doi.org/10.1093/bjps/axx023>
- Siegel, J. S., Power, J. D., Dubis, J. W., Vogel, A. C., Church, J. A., Schlaggar, B. L., & Petersen, S. E. (2014). Statistical improvements in functional magnetic resonance imaging analyses produced by censoring high-motion data points. *Human Brain Mapping*, 35(5), 1981–1996. <https://doi.org/10.1002/hbm.22307>
- Smith, D. V., Gseir, M., Speer, M. E., & Delgado, M. R. (2016). Toward a cumulative science of functional integration: A meta-analysis of psychophysiological interactions. *Human Brain Mapping*, 37(8), 2904–2917. <https://doi.org/10.1002/hbm.23216>
- Stroh, A.-L., Rösler, F., Dormal, G., Salden, U., Skotara, N., Hänel-Faulhaber, B., & Röder, B. (2019). Neural correlates of semantic and syntactic processing in German Sign Language. *NeuroImage*, 200, 231–241. <https://doi.org/10.1016/j.neuroimage.2019.06.025>
- Trimmel, K., van Graan, A. L., Caciagli, L., Haag, A., Koepp, M. J., Thompson, P. J., & Duncan, J. S. (2018). Left temporal lobe language network connectivity in temporal lobe epilepsy. *Brain*, 141(8), 2406–2418. <https://doi.org/10.1093/brain/awy164>

REVIEWERS' COMMENTS:

Reviewer #4 (Remarks to the Author):

I am a novel reviewer for this round of reviews, and thus am deliberately taking a broad rather than nit-picky view. I find the manuscript as a whole to be complex, but fairly logical and a valuable contribution to the field. The authors are fairly exhaustive in their use of multiple methods to explore this small sample. The use of the same task runs for all analyses inherently limits impact somewhat, as they come near to double-dipping statistically with the use of the same small dataset, and critically, the same data-driven ROI definition, over and over in different ways. However, I do not find it problematic per se, but rather opening an interesting question as to whether it should be surprising that univariate hypoactivation group differences would relate to PPI differences between those same regions, and DCM differences between those same regions (model 2), in the same people, given use of the same regions defined by univariate group hypoactivation, and the same BOLD data as input.

A few notes:

Methods: Age (range and mean) should be provided in the text of the Methods Subjects section (I know it is in Table 1). I found it surprising the authors braved a verbal response task, given known motion difficulties (especially in children), and found the motion data provided to Reviewer 3 in the rebuttal to be very reassuring (and maybe worth including in the supplemental). The response to R3 about the 25% voxels made sense for the PPI analysis (and the 10% results were indeed highly similar). However, it is then odd that they didn't use the same methodological choices for DCM, however, choosing to go with top 10% there (in their first model). Consistent logic helps the reader.

Power: The manuscript presents small groups considering the number of analyses of performed. Sample size and power limitations should be noted in the Limitations sections (which is currently missing).

Discussion: A Limitations section would be helpful, to note aspects that could be improved, limitations of interpretation (some noted above), and what next steps could address. One point that is also not highlighted in the discussion, that I think would be interesting, is that this is an adult sample, with many years of reading experience (presumably). Do some of the group differences reflect these years of experience/compensation/reading anxiety/difficulty (aka would not be seen in a developmental sample)?

Discussion model: The model (Fig 8) is a bit confusing to me, despite knowing it is stemming from the authors' DCM results. It has a flavor of the dorsal and ventral vision pathways, but the ventral/dorsal visual streams from the primate vision literature do not split as late as the vOTC, but rather starts earlier in the visual pathway. In the Coltheart dual route model for reading, both ventral and dorsal proceed simultaneously and in parallel from visual input, not from vOTC (ventral) to LTPC (dorsal). It is interesting to ask about communication flow between vOTC and TPC, and to think about the DCM results, but it is also highly likely that both locations are getting visual information from earlier vision areas as well, and I think that is important to note or be reflected in the figure, rather than the current Figure 8, which suggests that they are serial only.

Dear Reviewer 4,

Please find our detailed responses to your feedback (in *italics*) under the corresponding comment / paragraph below.

I am a novel reviewer for this round of reviews, and thus am deliberately taking a broad rather than nit-picky view. I find the manuscript as a whole to be complex, but fairly logical and a valuable contribution to the field. The authors are fairly exhaustive in their use of multiple methods to explore this small sample. The use of the same task runs for all analyses inherently limits impact somewhat, as they come near to double-dipping statistically with the use of the same small dataset, and critically, the same data-driven ROI definition, over and over in different ways. However, I do not find it problematic per se, but rather opening an interesting question as to whether it should be surprising that univariate hypoactivation group differences would relate to PPI differences between those same regions, and DCM differences between those same regions (model 2), in the same people, given use of the same regions defined by univariate group hypoactivation, and the same BOLD data as input.

A few notes:

Methods: Age (range and mean) should be provided in the text of the Methods Subjects section (I know it is in Table 1). I found it surprising the authors braved a verbal response task, given known motion difficulties (especially in children), and found the motion data provided to Reviewer 3 in the rebuttal to be very reassuring (and maybe worth including in the supplemental). The response to R3 about the 25% voxels made sense for the PPI analysis (and the 10% results were indeed highly similar). However, it is then odd that they didn't use the same methodological choices for DCM, however, choosing to go with top 10% there (in their first model). Consistent logic helps the reader.

We agree and have added age range and mean age in the Subjects section. We know that it is always a risk to have subjects speak in the scanner and we were indeed pleased that only few subjects, especially in the dyslexia group (many of whom had never been in an MRI), moved 'much'. We are happy to include the movement parameters in the supplements. To us, it was very important to make sure that subjects really read the stimuli and we could also look at reading accuracy. So many studies use lexical decision tasks but making a choice between a word and a pseudoword is so much easier and does not require the subject to really read what they see on the screen. Children with dyslexia guess a lot (at least from my own experience as dyslexia therapist) and I just wanted to make sure that what we capture with fMRI and the performance measures is really showing a reading process (or phonological decoding probably for complex pseudowords) and not just sight word reading judgments.

Power: The manuscript presents small groups considering the number of analyses of performed. Sample size and power limitations should be noted in the Limitations sections (which is currently missing).

While we agree that the sample size is not huge, the two groups were based on two neurostimulation studies, where sample sizes are usually smaller than in standard fMRI studies. However, we do understand your concerns and performed post-hoc sensitivity analyses to proof that we can draw our conclusions with some confidence. We have thus added the following section to the Methods section:

“ Since data from the present study originates from two neurostimulation studies, sample size was determined based on comparable previous neurostimulation studies and sensitivity analyses were performed to confirm the chosen sample size was sensitive enough to detect the expected effect size. Earlier neurostimulation studies that aimed to modulate reading performance in dyslexia had reported strong stimulation effects with Cohen's d ranging from -0.37 to 1.96 based on repeated measures ANOVAs with ten subjects^{60,61}. Similarly, work from our own group revealed strong stimulation effects after left TPC stimulation in a group of 26 healthy adult subjects (Cohen's $d = 0.63$)⁶². G-power was used to perform a sensitivity calculation, which reveals the smallest effect that could have been detected with high probability given our sample size. The sensitivity analysis showed that assuming $\alpha = 0.05$, we had 80% power to detect effect sizes larger than 0.55 (Cohen's d_z) for two-tailed t-tests and larger than 0.46 (Cohen's d) for repeated measures ANOVAs⁶³. ”

Discussion: A Limitations section would be helpful, to note aspects that could be improved, limitations of interpretation (some noted above), and what next steps could address. One point that is also not highlighted in the discussion, that I think would be interesting, is that this is an adult sample, with many years of reading experience (presumably). Do some of the group differences reflect these years of experience/compensation/reading anxiety/difficulty (aka would not be seen in a developmental sample)?

Thank you for pointing this out. We are happy to discuss a few of the limitations and show some future directions. We have added the following paragraph at the end of the Discussion section:

“One limitation of the present study is the difficulty of accounting for differences in reading training and compensation over the years, with each individual with dyslexia following their own trajectory of reading development and progress. The observed results might thus not only stem from general differences between typical and atypical readers, as found in children with dyslexia and typically reading peers, but also from compensatory mechanisms and processes influenced by extensive reading instruction, training and compensation strategies that developed over decades. While the sample size is rather small, we did our best to create a homogenous dyslexia group (e.g., by making sure that no subjects had attention or arithmetic deficits, met our stringent criteria for dyslexia status) and used standardized and manifold measures to assess reading abilities. We believe that the findings of the present study can guide future neurostimulation studies, which, for instance, could aim to tackle the observed hypoactivation in the right cerebellum or the left SMG with facilitatory stimulation protocols.”

Discussion model: The model (Fig 8) is a bit confusing to me, despite knowing it is stemming from the authors' DCM results. It has a flavor of the dorsal and ventral vision pathways, but the ventral/dorsal visual streams from the primate vision literature do not split as late as the vOTC, but rather starts earlier in the visual pathway. In the Coltheart dual route model for reading, both ventral and dorsal proceed simultaneously and in parallel from visual input, not from vOTC (ventral) to LTPC (dorsal). It is interesting to ask about communication flow between vOTC and TPC, and to think about the DCM results, but it is also highly likely that

both locations are getting visual information from earlier vision areas as well, and I think that is important to note or be reflected in the figure, rather than the current Figure 8, which suggests that they are serial only.

Thank you for pointing this out. Not coming from vision research per se, we had not really thought about the perception of our model. Our main aim was to show that we believe that the reading path follows a dorsal and not ventral route for both words and pseudowords in dyslexia. We agree that it is highly likely that both locations get visual input from vision areas but since we did not investigate this with our DCMs, we did not want to make any assumptions or draw conclusions. Therefore, we would rather not include the visual areas and emphasize in the figure caption that we are really only summarizing our DCM findings in relation to the dual-route of reading. We have therefore adapted it to the current form:

“Figure 8. Schematic illustration of processing along the dorsal reading route in dyslexia based on current DCM findings. Based on our DCM findings, we suggest that both words and pseudowords are at least partially processed via a dorsal route in dyslexia, but the engagement of TPC subregions might be crucial. Effective connectivity analyses suggest that the left vOTC to left pSTG (TPC) connection was modulated by words, whereas the connection to the left SMG was modulated by pseudowords in dyslexia. Our behavioural correlations support their implication for word and pseudoword reading. **Please note that the schematic illustration only displays the specific pathway via the dorsal route and does not imply an altered ventral route or serial processing.**”

Please note that we have also included motion parameters in the supplements (Supplementary Table 16):

Supplementary Table 16. fMRI motion parameters for all subjects, including the number of motion outliers and maximum (max_FD), mean (mean_FD) and standard deviation (std_FD) of framewise displacement (FD).

Subject	group	num_outliers	max_FD	mean_FD	std_FD
Sub-1	DYS	6	1.2959	0.2581	0.1651
Sub-2	DYS	2	0.9470	0.2917	0.1397
Sub-3	DYS	0	0.6794	0.1942	0.0995
Sub-4	DYS	39	5.9748	0.4150	0.3661
Sub-5	DYS	0	0.8007	0.2588	0.1209
Sub-6	DYS	0	0.7446	0.2026	0.1024
Sub-7	DYS	30	2.0691	0.4069	0.2718
Sub-8	DYS	59	1.7479	0.4661	0.2676
Sub-9	DYS	1	1.0999	0.1714	0.0939
Sub-10	DYS	0	0.6410	0.2076	0.1055
Sub-11	DYS	0	0.4630	0.1667	0.0693
Sub-12	DYS	0	0.4167	0.1490	0.0643
Sub-13	DYS	1	0.9234	0.2681	0.1265
Sub-14	DYS	1	0.9278	0.1920	0.1126
Sub-15	DYS	0	0.4910	0.1866	0.0849
Sub-16	DYS	1	1.0885	0.2065	0.1148
Sub-17	DYS	5	1.3238	0.3810	0.1902
Sub-18	DYS	21	6.1666	0.3328	0.3573

Sub-19	DYS	1	1.1901	0.2220	0.1190
Sub-20	DYS	15	2.9056	0.2339	0.2333
Sub-21	DYS	0	0.4390	0.1884	0.0856
Sub-22	DYS	1	0.9856	0.1970	0.1141
Sub-23	DYS	4	1.3790	0.2834	0.1580
Sub-24	DYS	6	2.6000	0.2080	0.1750
Sub-25	DYS	0	0.8364	0.1641	0.0882
Sub-26	DYS	0	0.8811	0.2253	0.1231
sub-1	CG	0	0.7690	0.2669	0.1273
sub-2	CG	11	1.6991	0.3850	0.1878
sub-3	CG	14	1.4774	0.3425	0.2075
sub-4	CG	2	0.9516	0.2716	0.1602
sub-5	CG	0	0.5382	0.1706	0.0761
sub-6	CG	19	1.5951	0.3422	0.2124
sub-7	CG	0	0.5043	0.2006	0.0832
sub-8	CG	0	0.4668	0.1114	0.0553
sub-9	CG	0	0.8473	0.1644	0.1105
sub-10	CG	0	0.7335	0.2163	0.0978
sub-11	CG	0	0.8504	0.2956	0.1499
sub-12	CG	6	1.8085	0.2695	0.1742
sub-13	CG	0	0.8594	0.2215	0.1325
sub-14	CG	0	0.6814	0.1891	0.0878
sub-15	CG	0	0.8589	0.1680	0.1033
sub-16	CG	0	0.6940	0.1516	0.0862
sub-17	CG	3	1.2210	0.3112	0.1507
sub-18	CG	0	0.8289	0.2679	0.1311
sub-19	CG	0	0.4869	0.1497	0.0793
sub-20	CG	2	2.3739	0.1669	0.1447
sub-21	CG	78	8.3539	0.5476	0.8127
sub-22	CG	0	0.4749	0.1285	0.0703
sub-23	CG	6	1.2003	0.2565	0.1602
sub-24	CG	0	0.5380	0.1234	0.0661
sub-25	CG	1	0.9580	0.2284	0.1209
sub-26	CG	0	0.7583	0.1890	0.1023
sub-27	CG	8	1.4499	0.2650	0.1870
sub-28	CG	0	0.7833	0.2106	0.1129